

# Development of a plant carbon-nitrogen interface coupling framework in a coupled biophysical-ecosystem-biogeochemical model (SSiB5/Triffid/DayCent-SOM v1.0): Its parameterization, implementation, and evaluation

Zheng Xiang[1,2], Yongkang Xue[2*], Weidong Guo[1,4*], Melannie D. Hartman[3], Ye Liu[2,5], William J. Parton[3]

[1]School of Atmospheric Sciences, Nanjing University, Nanjing, China
[2]Department of Geography, University of California, Los Angeles, CA 90095, USA
[3]Natural Resource Ecology Laboratory, Colorado State University, CO 80523, USA
[4]Joint International Research Laboratory of Atmospheric and Earth System Sciences, Nanjing, China
[5]Pacific Northwest National Laboratory, Richland, WA 99352, USA.

*Correspondence to*: Yongkang Xue (yxue@geog.ucla.edu), Weidong Guo (guowd@nju.edu.cn)

**Abstract.** Plant and microbial nitrogen (N) dynamics and N availability regulate the photosynthetic capacity and capture, allocation, and turnover of carbon (C) in terrestrial ecosystems. Studies have shown that a wide divergence in representations of N dynamics in land surface models leads to large uncertainty in the biogeochemical cycle of the terrestrial ecosystems and then in climate simulations as well as the projections of future trajectories. In this study, a plant C-N interface coupling framework is developed and implemented in a coupled biophysical-ecosystem-biogeochemical model (SSiB5/TRIFFID/ 20  DayCent-SOM v1.0). The main concept and structure of this plant C-N framework and its coupling strategy are presented. This framework takes more plant N-related metabolism processes into account. For instance, plant resistance and self-adjustment is represented by a dynamic C/N ratio for each plant functional type (PFT). Furthermore, when available N is less than plant N demand, plant growth is restricted by a lower maximum carboxylation capacity of Rubisco (Vmax) level, reducing gross primary productivity (GPP). In addition, a module for plant respiration rates is introduced by adjusting the respiration 25  with different rates at different plant components for the same N concentration. Since insufficient N can potentially give rise to lags on plant phenology, phenology scheme is also adjusted with a lag factor related to N processes. All these considerations ensure a more comprehensive incorporation of N regulations to plant growth and C cycling. This new approach has been tested systematically to assess the effects of this coupling framework and N limitation on the terrestrial carbon cycle. Long term measurements from both flux tower sites with different PFTs and global satellite-derived products are employed as references 30  to assess these effects. The results show a general improvement with the new plant C-N coupling framework, with more



consistent emergent properties, such as GPP and leaf area index (LAI), compared to observations. The main improvements occur in tropical Africa and boreal regions, accompanied by a decrease of the bias in global GPP and LAI by 16.3% and 27.1%, respectively.

## 1 Introduction

Land surface processes substantially affect climate (Foley et al., 1998; Ma et al., 2013; Sellers et al., 1986; Xue et al., 2004, 2010, 2022, 2023) and are influenced by climate in turn (Bonan, 2008; Liu et al., 2019, 2020; Zhang et al., 2015), forming complex feedback loops to climate change (Friedlingstein et al., 2006; Gregory et al., 2009). To study these processes, the land surface components of Earth System Models (ESMs) have evolved from those that only represent biophysical processes (i.e., hydrology and energy cycle) to those that include the terrestrial carbon (C) cycle, vegetation dynamics, and nutrient processes

(Cox, 2001; Dan et al., 2020; Foley et al., 1998; Jiang et al., 2014; Niu et al., 2020; Oleson et al., 2013; Pan et al., 2017; Sellers et al., 1986; Sitch et al., 2003; Wang et al., 2010; Yang et al., 2019; Zhan et al., 2003).

Current land surface models have large uncertainties in predicting historical and recent C exchanges (Beer et al., 2010; Richardson et al., 2012; Zaehle et al., 2015). The parameterization of some processes has been criticized for being oversimplified from an ecological point of view (Ali et al., 2015; Lawrence et al., 2019; Reich et al., 2006), and the dynamic

vegetation models tend to overestimate terrestrial C sequestration (Anav et al., 2015; Murray-Tortarolo et al., 2013; Mueller et al., 2019; Gristina et al., 2020; Oliveira et al., 2021; Heikkinen et al., 2021).The uncertainty/errors in predictions using land models have been attributed to many factors. The inclusion or exclusion of nutrient limitations on productivity is one of the critical factors. Those C-only models ignore significant nitrogen (N) impacts and therefore overestimate C sequestration by terrestrial ecosystems under climate change (Peñuelas et al., 2013; Zaehle et al., 2015). Ecosystem N cycling processes are

among the dominant drivers of terrestrial C-climate interactions through their impacts, mainly N limitation, on vegetation growth and productivity (Reich et al., 2006), especially in N-poor younger soils at high latitudes (LeBauer & Treseder, 2008; Vitousek and Howarth, 1991), and on microbial decomposition of organic matter (Hu et al., 2001). As such, the N cycle and its effect on C uptake in the terrestrial biosphere has been incorporated in land surface models (LSMs) of ESMs (Davies-Barnard et al., 2020)with various representations of N processes (Ali et al., 2015; Asaadi et al., 2021; Best et al., 2011; Clark

et al., 2011; Davies-Barnard et al., 2020; Ghimire et al., 2016; Goll et al., 2017; Krinner et al., 2005; Lawrence et al., 2019; Matson et al., 2002; Oleson et al., 2013; Smith et al., 2014; Thum et al., 2019; Wang et al., 2010; Wiltshire et al., 2020; Yu et al., 2020; Zhu et al., 2019).

In the latest Coupled Model Intercomparison Project Phase 6 (CMIP6, Eyring et al., 2016), although there were 112 different coupled models with various land surface models from 33 research teams, only about 10 models incorporated an N cycle

module (Arora et al., 2020). The coupling of N processes is still an area of model development. Among these models including N processes, most of them pay more attention to microbial N dynamics in soil. The adequate C-N coupling in plant N processes has been indicated as an area that still needs intensive investigation (Thum et al., 2019; Ghimire et al., 2016; Goll et al., 2017;




Yu et al., 2020; Zaehle et al., 2015; Zhu et al., 2019). Some key plant N processes, such as N limitation on GPP, the effect of biomass N content on autotrophic respiration, plant N uptake, ecosystem N loss, and biological N fixation, have been

introduced into LSMs with various complexity to present the N limitation effects in current land models. They include, for instance, using N to scale down the photosynthesis parameter $V_{c,max}$ (Ghimire et al., 2016; Zaehle et al., 2015) or potential GPP to reflect N availability (Gerber et al., 2010; Oleson et al., 2013; Wang et al., 2010); defining a C cost of N uptake (Fisher et al., 2010) and optimizing N allocation for leaf processes (Ali et al., 2015). In many of these approaches, N limitation is represented as instantaneous down-regulation of potential photosynthesis rates based on soil mineral N availability.

This paper presents a recently developed process-based approach, which mainly focuses on the N limitation effects and plant resistance on photosynthesis, plant respiration, and plant phenology. The dynamic plant C/N ratio is a key concept in representing plant resistance, self-adjustment, and C/N interactions. Due to their relative immobility, plants often face significant challenges in obtaining an adequate supply of nutrients to meet the demands of basic cellular processes. A deficiency of any type of nutrient may result in decreased plant productivity and/or fertility (McDowell et al., 2008; Morgan

and Connolly, 2013; Stenberg and Muola, 2017). Evidence has shown that plant C/N ratios have to change over the plant's lifecycle with nutrient availability (Chen and Chen, 2021; McGroddy et al., 2004; Meyer-Grünefeldt et al., 2015; Sardans et al., 2012; Smith, 1991; Yang et al., 2021; Zhang et al., 2011) through plant self-adjustment. Plant cells C/N ratios are influenced by the accumulation of C polymers, such as carbohydrates and lipids, and are greater when cells are nutrient starved, or exposed to high light (Aber et al., 2003; MacDonald et al., 2002; Talmy et al., 2014). However, many land models specify fixed plant

C/N ratios for each plant functional type (PFT) (e.g., Best et al., 2011; Clark et al., 2011; Krinner et al., 2005; Oleson et al., 2013; Wang et al., 2010). In this paper, we present a new plant C-N coupling framework with flexible C/N ratios (Section 2.2.2), in which N regulates photosynthesis (Section 2.2.3), respiration (Section 2.2.4), and plant phenology (Section 2.2.5), as well as produces a consistent coupling between biophysical and biogeochemical processes. Allometric relations and empirical data sets are used to constrain the range of possible C/N ratios. This dynamic C/N ratio depends on the degree to

which the N demands of different plant organs (e.g., leaf, root, and wood) are satisfied over the past several days. This plant C-N framework takes some plant N metabolism processes into account and prevents unrealistic instantaneous down-regulation of potential photosynthesis rates.

We implement this plant C-N framework by coupling a soil organic matter and nutrient cycling model (DayCent-SOM, Del Grosso et al., 2000; Parton et al., 1998, 2010)with a biophysical/dynamic vegetation model (SSiB5/TRIFFID, the Simplified

Simple Biosphere Model version 5/ Top-down Representation of Interactive Foliage and Flora Including Dynamics Model, Cox, 2001; Harper et al., 2016; Liu et al., 2019; Xue et al., 1991; Zhan et al., 2003; Zhang et al., 2015). DayCent-SOM, which includes only the soil organic matter (SOM) cycling and trace gas subroutines from the DayCent ecosystem model (Parton et al., 1998, 2010) represents SOM transformations, below-ground N cycling, soil N limitation to microbial processes and plant growth, and nitrification/denitrification processes. In the coupled model, the potential N uptake depends on plant N demand

from a biophysical and dynamic vegetation model, SSiB5/TRIFFID. The actual plant N uptake is limited based on soil N availability as predicted by DayCent-SOM ( Del Grosso et al., 2000; Parton et al., 1998; 2010). The coupled model is verified



at twelve flux tower sites with different PFTs and is used to conduct several sets of global 2-D offline simulations from 1948 to 2007 to assess the effects of the coupling process. The model predictions of global GPP and LAI are evaluated against satellite-derived observational data. The results demonstrate the relative importance of different plant N processes in this C-N framework.

The model used in this paper is presented in section 2.1. The development and implementation of this plant C-N framework is presented in section 2.2. The model forcing and validation data used in this paper are presented in section 2.3. In section 3, the experimental design is described. In section 4, the measurements from the flux tower sites with different PFTs and the global satellite-derived observations from 1948-2007 are used as references to assess the effect of the C-N coupling process on the long-term mean vegetation distribution and terrestrial C cycling using the offline SSiB5/TRIFFID/DayCent-SOM. Some issues and conclusions are presented in section 5.

## 2 Methods

### 2.1 Model description

#### 2.1.1 SSiB4/TRIFFID model

The Simplified Simple Biosphere Model (SSiB, Xue et al., 1991; Sun and Xue, 2001; Zhan et al., 2003) is a biophysical model that simulates fluxes of surface radiation, momentum, sensible/latent heat, runoff, soil moisture and temperature, and vegetation GPP based on energy and water balance and photosynthesis processes. The SSiB was coupled with a dynamic vegetation model, the Top-down Representation of Interactive Foliage and Flora Including Dynamics Model (TRIFFID), to calculate NPP, LAI, canopy height, and PFT fractional coverage according to the C balance (Cox, 2001; Harper et al., 2016; Liu et al., 2019; Zhang et al., 2015). Moreover, the surface albedo and aerodynamic resistances are also updated based on the vegetation conditions. Previous work has improved the PFT competition strategy and plant physiology processes to make the SSiB4/TRIFFID suitable for seasonal, interannual, and decadal studies (Zhang et al., 2015). SSiB4/TRIFFID includes seven PFTs: (1) broadleaf evergreen trees (BET), (2) needleleaf evergreen trees (NET), (3) broadleaf deciduous trees (BDT), (4) C3 grasses, (5) C4 plants, (6) shrubs, and (7) tundra. PFT coverage is determined by net C availability, competition between species, and disturbance, which includes mortality due to fires, pests, and windthrow. A detailed description and validation of SSiB4/TRIFFID can be found in Zhang et al. (2015), Liu et al. (2019), and Huang et al. (2020). In this study, The DayCent-SOM (see next section) is introduced and coupled with the SSiB5/TRIFFID using the C-N interface coupling framework introduced in this study, which will be discussed in Section 2.2.

#### 2.1.2 DayCent-SOM model

DayCent-SOM, a subset of DayCent that excludes plant growth, soil hydrology, and soil temperature subroutines, consists of soil mineral N pools (ammonium and nitrate) and six types of organic C and N pools consisting of two non-woody plant litter



pools (metabolic and structural), three coarse woody debris pools (from death of large wood, fine branches, and coarse roots), and three kinetically defined organic matter pools, (active, slow, and passive); all types of organic pools except the passive pool have both above-ground and below-ground counterparts. Non-woody plant litter is partitioned into structural (lignin + cellulose) and metabolic (labile) litter based on the lignin: N ratio of plant material (Parton et al., 1994)). The coarse woody debris pools decay in the same way that the structural pool decomposes, with lignin and cellulose going to the slow soil organic matter pool and the labile fraction going to the active soil organic matter pool. Each type of organic pool has its own intrinsic rate of decomposition, modified by temperature and moisture effects (Parton et al., 1994). Additionally, the decomposition rates of the structural material and coarse woody debris pools are functions of their respective lignin fractions. DayCent's litter decay model has been validated using extensive data from the LIDET litter decay experiments from all over the world(Bonan et al., 2013)

## 2.2 Development of a plant Carbon-Nitrogen (C-N) Interface coupling framework

### 2.2.1 Conceptual considerations and coupling strategy

To represent C/N interactions, we have developed a plant C-N interface framework to take into account both biophysical and biochemical C/N processes in plant life activities. In this study, we applied the coupling framework to SSiB5/TRIFFID/ DayCent-SOM. However, this approach could be applied to any other process-based models with similar physical/biological principles. The conceptual considerations in developing this framework are presented in this section. For a process-based model, introducing a consistent coupling philosophy between biophysical and biogeochemical processes is necessary. The soil N dynamics model (DayCent-SOM) is directly driven by soil temperature/moisture as well as plant C/N litter inputs into soil. Because the surface water, radiation, and carbon fluxes and plant litter are calculated by SSiB5, we force DayCent-SOM with SSiB5-produced soil temperature, soil moisture, and SSiB5/TRIFFID-produced plant litter. DayCent-SOM then computes daily changes of all organic matter and mineral soil pools, estimates losses of N from nitrate leaching and $N_2O$, $NO_x$, and $N_2$ emissions, predicts the amount of inorganic N available to plants, and updates inorganic N pools after accounting for plant N uptake by SSiB5. Following plant N uptake from DayCent-SOM, our plant C-N interface framework describes N effects on plant physiology from photosynthesis, plant autotrophic respiration, and plant phenology plus a dynamic C/N ratio (Fig. 1). Following such model development philosophy, our framework not only considers N limitation on the general plant photosynthesis process but also emphasizes the N limitation effect during the growth season, which can represent the physiological processes of C-N cycling and help us obtain more information to understand the attribution of N processes on the C cycle.



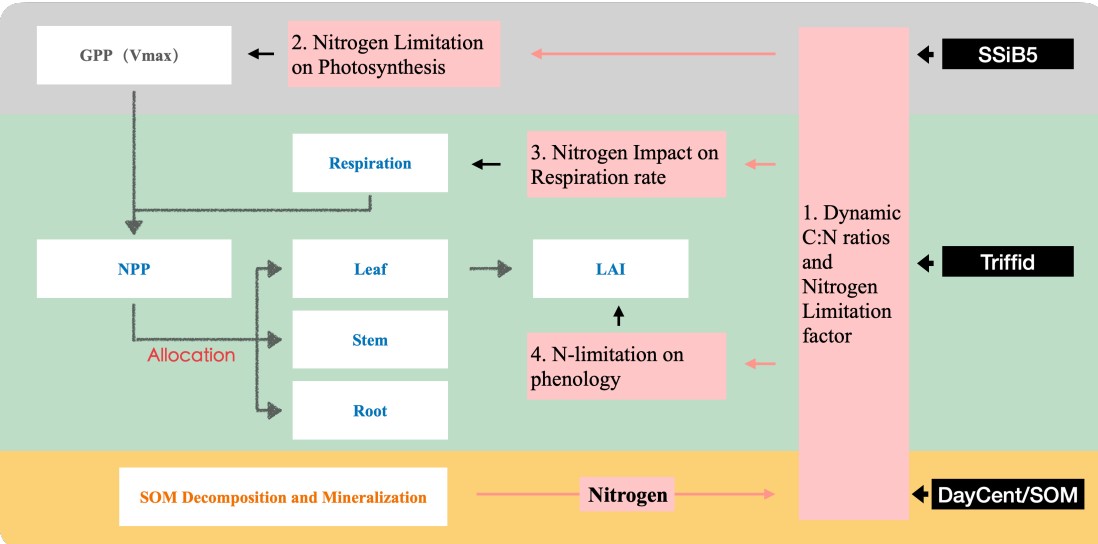


**Figure 1.** Schematic diagram of plant biogeochemistry and nitrogen impacts in SSiB5/TRIFFID/DayCent-SOM.

Notes: (1) Different background color represents three different modules: SSiB, TRIFFID, and DayCent/SOM; (2) White boxes indicate the main processes in C-N coupling in different modules; (3) Vermeil boxes indicate how nitrogen influences plant biogeochemistry through

the C-N framework.

In the original land surface model (SSiB4/TRIFFID), with assumed unlimited N availability and fixed C/N ratios based on PFT, the assimilated C determined the N contents of leaf, stem, and root, which influenced autotrophic respiration, followed by GPP, LAI, and NPP. However, more evidence indicates the C/N ratio is not fixed in plant life. The studies of Ecological

Stoichiometry (Sterner and Elser, 2002), which investigates how the availability of multiple elements, including carbon, nitrogen, and phosphorus, constrain ecological interactions, reveals that plants can adjust their resource requirements. Changes in N resource availability will result in changes to plant C allocation and partitioning. Studies show that plants resorb only about 50% of leaf N on average (Aerts, 1996) to conserve nutrients (Clarkson and Hanson, 1980) and to increase nutrient use efficiency (Herbert & Fownes, 1999; Vitousek, 1982). These processes cause a major internal nutrient flux and changes of

C/N ratios to reduce the impact of N limitation (Talhelm et al., 2011; Vicca et al., 2012). In addition, plant responses, such as plant resistance and self-adjustment, will be limited under fixed C/N ratios, which affect plant productivity and litter N content, thus driving changes in the underground biogeochemistry and ultimately C and N uptake and storage (Drewniak and Gonzalez-Meler, 2017). Some studies show that the increase in foliar N under increased soil N availability would improve plant responses because it allows adaptations in the stoichiometry of C and N (Bai et al., 2021; Chang et al., 2022; Ding et al., 2022; Kaiser et

al., 2014; Lin et al., 2022; Zhang et al., 2022) . The main impact will be to decrease C/N ratio in leaves when the available N increases, driving increases in productivity and changes in soil and litter N content (Drewniak and Gonzalez-Meler, 2017). A dynamic C/N ratio is employed in our framework to obtain N states more realistically and properly represent the effect of N processes (See section 2.2.2 for more details).



A commonly used parameterization of photosynthetic C assimilation by the terrestrial biosphere in ESMs is represented by the
Farquhar, von Caemmerer, and Berry (FvCB) model of photosynthesis (Collatz et al., 1991; Farquhar et al., 1980). Plants
require N as essential components of photosynthetic proteins involved in light capture, electron transport, and carboxylation
(Evans, 1989). Nitrogen is an important constituent of the Rubisco enzyme and mitochondrial enzymes that regulate respiration
and adenosine triphosphate (ATP) generation (Makino and Osmond, 1991). One of the most important photosynthetic model
parameters, the maximum carboxylation rate by the Rubisco enzyme ($V_{c,max}$) is a key parameter in the FvCB model (Farquhar
et al., 1980) and has an extensive range across the models depending on the plant N content (Rogers, 2014). Since N is an
important component of the Rubisco enzyme, leaf N content will affect $V_{c,max}$ and thus GPP. The original FvCB model did
not explicitly consider the N effect on photosynthesis. In a number of LSMs, an empirical relationship is applied to relate
$V_{c,max}$ to leaf N content $N_{leaf}$ to generate the effect of N on photosynthesis, e.g., ,$V_{c,max} = i_v + s_v \times N_{leaf}$, where the intercept
($i_v$) and slope ($s_v$) are derived for each PFT based on observations (Kattge et al., 2009; Raddatz et al., 2007). Moreover, in
some coupling approaches, only the relationship between the root N uptake and GPP/NPP is considered to represent the N
limitation on C cycles (Ali et al., 2015; Fisher et al., 2010; Ghimire et al., 2016). However, because NPP is the difference
between GPP and autotrophic respiration, adjusting NPP or GPP alone may cause the ratio between NPP and respiration to
deviate from reality. Using the process-based N limitation factor produced from DayCent-SOM to modify $V_{c,max}$ (See section
2.2.3) is a more realistic way to produce the N effect on the photosynthesis process. Moreover, when calculating the N limit
to Rubisco capacity, an aforementioned dynamic C/N ratio is introduced in our approach to influence the N-limit effect on
photosynthesis.

Nitrogen is not only a dominant regulator of vegetation dynamics, GPP, NPP, and terrestrial C cycles; Reich et al., (2008)
demonstrate strong relationships between respiration and N limitation based on observational data from various species. At
any normal N concentration, respiration rates are consistently lower on average in leaves than in stems or roots. Therefore, we
introduce two parameters for stems and roots, respectively, based on PFT to adjust the respiration rate in section 2.2.4.

Nitrogen also affects plant phenology and can be remobilized to supply spring bud-break or vegetative shoot extension (Kolb
and Evans, 2002; Marmann et al., 1997; Millard, 1994; Neilsen et al., 1997). Nitrogen resorption is found during leaf
senescence and growth in evergreens (May and Killingbeck, 1992). Because plants need time to turnover, the plant N processes
also have a lag effect on plant phenology (Thomas et al., 2015). Phenology in SSiB4/TRIFFID modulates LAI evolution,
including leaf mortality, but it is not directly linked to N. Since different N states and supplements will lead to different lags
on phenology, we add N impact on plant phenology by introducing a N limitation parameter, which will be discussed in section
2.2.5.

### 2.2.2 Dynamic C/N ratio based on plant growth and soil nitrogen storage

Plant resistance and self-adjustment to N availability ($N_{avail}$) are represented by dynamic C/N ratios (CNRs) in SSiB5. The N
availability for new growth limits the C assimilation rate through the CNRs, i.e., the model simulated NPP should be no more





than $N_{avail} \times$ CNR of new plant material. In the original TRIFFID parameterization, the CNRs for different plant components (leaf, root, and wood) are fixed based on plant functional types (Cox, 2001); and changes in CNR occurred over the lifecycle of the plant, varied with nutrient availability, and were not considered in the original SSiB4/TRIFFID models. A linear relationship between CNR and $N_{avail}$, based on the DayCent's parameterization, is introduced to the

SSiB5/TRIFFID/DayCent-SOM for each PFT's components, (Fig. 2, Eq. 1).

$$CNR = \begin{cases} CNR_{max}, & N_{avail} \leq N_{min} \\ \frac{N_{avail} - N_{,max}}{N_{min} - N_{max}} \times CNR_{min} + \frac{N_{avail} - N_{min}}{N_{max} - N_{min}} \times CNR_{max} & N_{min} < N_{avail} < N_{max} \\ CNR_{min}, & _{avail} \geq N_{max} \end{cases} \quad (1)$$

where $N_{avail}$ is the amount of soil mineral nitrogen that was available at the end of the previous day (g N m$^{-2}$) calculated from DayCent-SOM.

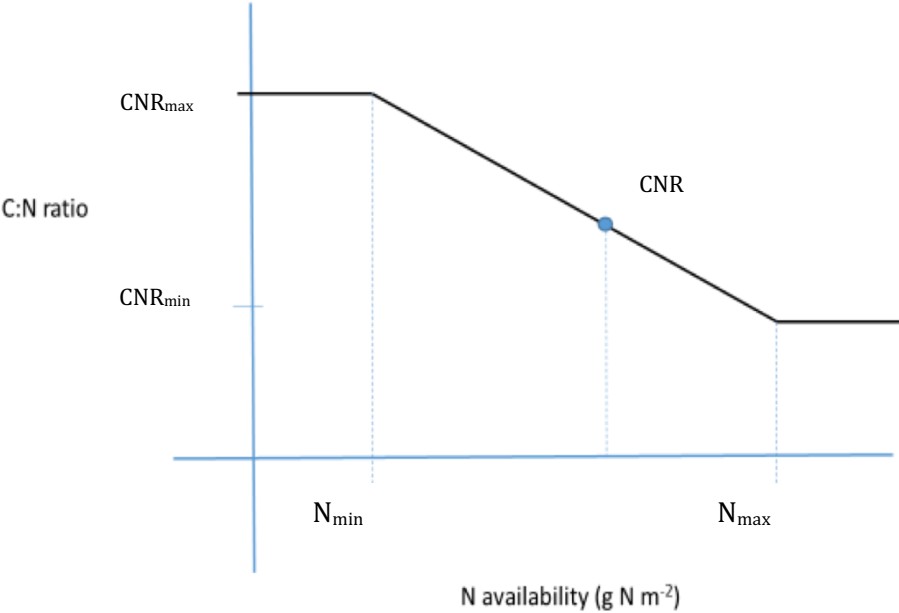

**Figure 2.** The relationship between soil nitrogen availability and plant carbon-nitrogen ratios.

The minimum and maximum amounts of nitrogen ($N_{min}$, $N_{max}$) necessary for the potential $NPP_p$ (g C m$^{-2}$ day$^{-1}$), which is first calculated from the SSiB4/TRIFFID with unlimited N, are:

$$N_{min} = \frac{NPP_p}{CNR_{max}} \quad (2)$$

$$N_{max} = \frac{NPP_p}{CNR_{min}} \quad (3)$$

where $CNR_{min}$ and $CNR_{max}$ are the minimum and the maximum C/N ratio for each PFT's components from the DayCent (Table 1). The CNR ranges of leaves, fine roots, and stems/wood are from DayCent's user's manual and other published papers





(Parton et al., 1993, 2007). Please note that Eq. (2) and Eq. (3) are calculated based on potential NPP; the CNR that is calculated based on Eqs. 1-3 ensures that when $N_{avail}$ varies between $N_{min}$ and $N_{max}$, the plant can adjust CNR to support this potential

NPP (as demonstrated in the schematic diagram Figure 2). That said, the N limitation will have no effect on the C assimilation as long as $N_{avail}$ is larger than $N_{min}$. However, the N content of plant litter falling to the soil was determined by this dynamic C/N ratio. Compared with the constant CNR, the range of possible plant carbon variation with dynamic CNR is smaller, reducing the impact of N limitation. As reviewed in the introduction and previous section, a number of recent studies have demonstrated that allowing adaptations in the stoichiometry of C and N would improve plant responses; for instance, an

increase in available foliar N decreases the C/N ratio in leaves, driving an increase in productivity.

**Table 1.** C:N ranges of leaves, fine roots, and stems/wood for each plant function type (PFT).

|  | Plant component | $CNR_{min}$ | $CNR_{max}$ |
|---|---|---|---|
| Broadleaf deciduous | leaves | 20 | 50 |
|  | roots | 40 | 70 |
|  | wood | 200 | 500 |
| Broadleaf Evergreen | leaves | 20 | 40 |
|  | roots | 40 | 70 |
|  | wood | 150 | 300 |
| Needleleaf Evergreen | leaves | 30 | 60 |
|  | roots | 40 | 60 |
|  | wood | 400 | 800 |
| C3 grass | leaves | 20 | 40 |
|  | roots | 40 | 50 |
|  | wood | 40 | 80 |
| C4 grass | leaves | 20 | 60 |
|  | roots | 60 | 100 |
|  | wood | 60 | 100 |
| shrub | leaves | 20 | 40 |
|  | roots | 40 | 70 |
|  | wood | 200 | 400 |
| tundra shrub | leaves | 20 | 40 |
|  | roots | 40 | 80 |
|  | wood | 300 | 700 |

Note: Data of $CNR_{min}$ and $CNR_{max}$ for each PFT's components are from DayCent's user's manual and other publications

(Parton et al., 1993, 2007)





The DayCent-SOM only provides the total available nitrogen ($N_{avail}$) for the plant within one grid box, which consists of several PFTs. To apply equation 1, the nitrogen available for each PFT and its plant components in the grid box is calculated as

$N_{avail}(\text{i}) = N_{avail} * frac_i$                                                                          (4)

$N_{avail}(\text{i}, \text{j}) = N_{avail}(\text{i}) * \Delta C_j / \Sigma_j \, \Delta C_j$                                                          (5)

where $frac_i$ is the fraction of PFT i in one grid, and $\Delta C_j$ is the fraction of carbon allocated to plant component j, which consists of leaf, root, and wood, and is calculated in TRIFFID.

### 2.2.3 Effect of nitrogen limitation on photosynthesis based on soil available nitrogen and plant C-N ratio

There are several different ways to represent N limitation effects in land models, including using N to scale down photosynthesis (Ghimire et al., 2016; Goll et al., 2017; Thum et al., 2019; Yu et al., 2020; Zaehle et al., 2015; Zhu et al., 2019), scaling down potential GPP based on N availability (Gerber et al., 2010; Oleson et al., 2013; Wang et al., 2010), or defining an NPP cost of nitrogen uptake (Fisher et al., 2010). We choose the most physiological way by adjusting maximum Rubisco carboxylation rate ($V_{c,\max}$) during the photosynthesis process, rather than adjusting NPP at the end of the photosynthesis

process. $V_{c,max}$ regulates both C assimilation and autotrophic respiration; and the photosynthesis assimilation product, GPP, is proportional to $V_{c,max}$, which is proportional to nitrogen content of the Rubisco leaf reserves.

We therefore introduce a downregulation of the canopy photosynthetic rate based on the available mineral N for new growth ($N_{avail}$) using a N-availability factor, $f(N)$.

$V_{c,max,Nlimit} = V_{c,max} * f(N)$                                                               (6)

The $f(N)$ is determined by nitrogen availability:

$$f(N) = \begin{cases} \dfrac{N_{avail}}{N_{min}} & N_{avail} \le N_{min} \\ 1 & otherwise \end{cases} \qquad\qquad (7)$$

Because plants can adjust the relative allocations of C and N during N uptake via N remobilization and resorption to reduce the impact of N limitation, as discussed in the previous section for dynamic CNR, the N-limit effect on photosynthesis only applies when nitrogen availability is lower than the minimum amounts of nitrogen ($N_{min}$) necessary for the potential $NPP$.

We take into account that plants have resistance and self-adjustment through this approach to make the N-limit effect neither linearly nor instantaneously responsive to available N content. A linear relationship between $f(N)$ and $N_{avail}$ is valid only when N availability is not sufficient for the minimum N demand for new growth.

In fact, the factor, $f(N)$ can also be applied to NPP and GPP as shown in Equations 8a –b.

$NPP_{Nlimit} = NPP * f(N)$                                                               (8a)

$GPP_{Nlimit} = GPP * f(N)$                                                              (8b)

If NPP is adjusted (Eq. 8a), this means the same N limitation for photosynthesis is applied for plant respiration, which is not reasonable based on plant physiology (Högberg et al., 2017). Such approach may distort the ratio of NPP and respiration. If





only GPP is adjusted for N-limitation, then the N limitation for respiration is ignored. Therefore, we choose an approach to adjust $V_{c,max}$, which is related to N during the photosynthesis process and affects both C uptake and autotrophic respiration.

### 2.2.4 Improvement of nitrogen impact on respiration rates based on field observations

Nitrogen affects plant respiration (Reich et al., 2008; Thornely and Johnson, 1990). In the original SSiB4/TRIFFID, the total maintenance respiration ($R_{pm}$) is given by Cox (2001):

$$R_{pm} = 0.012 R_{dc} \frac{N_l + N_s + N_r}{N_l} \tag{9}$$

where $R_{dc}$ is canopy dark respiration and is linearly dependent on $V_{c,max}$. The introduced N limitation of $V_{c,max}$ in Section 2.2.3 also influences the N effect on maintenance respiration. $N_l$, $N_s$ and $N_r$ are the N contents of leaf, stem, and root, respectively, and the factor of 0.012 is from the unit conversion. Eq. (9) assumes the respiration rates in root and stem have the same dependence on N content as leaf. However, studies (Reich et al., 2008) have shown that the respiration rates at any common N concentration were consistently lower in leaves than in stems or roots on average. Thus, we introduce two PFT-specific parameters ($ResA_S$, $ResA_R$) from field observations (Wang et al., 2006; Yang et al., 1992) to adjust root and stem respirations. Their values are listed in Table 2.

$$R_{pm,Nlimit} = 0.012 R_{dc} \frac{N_l + ResA_S * N_s + ResA_R * N_r}{N_l} \tag{10}$$

**Table 2**. The values of $ResA_S$ and $ResA_R$ for each plant function type (PFT).

| PFT | Broadleaf deciduous | Broadleaf Evergreen | Needleleaf Evergreen | C3 grass | C4 grass | shrub | tundra shrub |
|---|---|---|---|---|---|---|---|
| $ResA_S$ | 1.36 | 1.36 | 1.44 | 1.0 | 1.0 | 1.25 | 1.25 |
| $ResA_R$ | 1.72 | 1.72 | 1.95 | 1.3 | 1.3 | 1.40 | 1.40 |

Since $ResA_S$ and $ResA_R$ are generally larger than 1, new $R_{pm}$ is larger than the original one, and the increased respiration due to the nitrogen limitation will decrease the NPP.

### 2.2.5 N limitation on LAI based on plant phenology

Studies (Aerts and Berendse, 1988; Thomas et al., 2015) show that leaf turnover and aboveground productivity are related to nutrient availability and that plant N processes can potentially give rise to lags on phenology. In TRIFFID, a leaf phenology parameter, $p$, (Cox, 2001) is introduced to represent the vegetation's phenological status, to calculate the leaf drop rate, and to adjust the model-simulated maximum possible LAI, which is based on carbon balance, ($LAI_{balance}$), to actual LAI and produce realistic phenology.





$$LAI = p \times LAI_{balance} \tag{11}$$

and

$$\frac{dp}{dt} = \begin{cases} -\gamma_p & \gamma_{lm} > 2\gamma_0 \\ \gamma_p(1-p) & \gamma_{lm} \leq 2\gamma_0 \end{cases} \tag{12}$$

where leaf constant absolute drop rate $\gamma_p = 20 \, yr^{-1}$, the leaf mortality rate $\gamma_{lm}$ is a function of temperature $T$ (Cox, 2001), and the minimum leaf turnover rate $\gamma_0 = 0.25$ (Cox, 2001). This phenology parameter, $p$, indicates that "full leaf" is approached asymptotically during the growing season, and $p$ is reduced at a constant absolute rate when the mortality rate is larger than a threshold value. Otherwise, $p$ increases but the rate of increase is reduced as the growing season evolves. To

reflect the N limitation in SSiB5/TRIFFID/DayCent-SOM, we assume $p$ is limited by N availability with the new nitrogen limited $p_{N \, Limit}$ determined by

$$p_{N \, Limit} = f(N) \times p \tag{13}$$

where $f(N)$ is calculated in section 2.2.3.

### 2.2.6 The computational flow of SSiB5/TRIFFID/DayCent-SOM

In SSiB5/TRIFFID/DayCent-SOM, SSiB5 provides GPP, autotrophic respiration, and other physical variables such as canopy and soil temperatures and soil moisture every 3 hours for TRIFFID (Fig. 3). TRIFFID accumulates the GPP from SSiB5 and produces biotic C, PFT fractional coverage, vegetation height, and LAI every ten days, which are used to update surface properties in SSiB5, such as albedo, roughness length, and aerodynamic/canopy resistances. The plant C-N framework uses the meteorological forcings (i.e., air temperature, humidity, wind, radiation, and precipitation) and physical variables (i.e., soil

moisture and soil temperature) provided by SSiB5 every 3 hours and the biophysical properties (vegetation fraction and biotic C) provided by TRIFFID, which is updated every ten days. The plant C-N interface framework calculates dynamic C/N ratios, N-limited photosynthesis, and N-impacted respiration rate every 3 hours. The C loss and potential N uptake are accumulated within one day in the C-N Interface Framework and plant C and N litter fall are transferred to DayCent-SOM at the end of the day. DayCent-SOM calculates inorganic N available for plant N uptake ($N_{avail}$) and N losses from nitrate leaching and N-

trace gas emissions each day. TRIFFID updates the vegetation dynamics based on C balance on day 10, including PFT competition. The updated vegetation dynamics are transferred to SSiB5 to calculate N-limited phenology to reflect N impact on the C cycle, which is significant during the growth season.



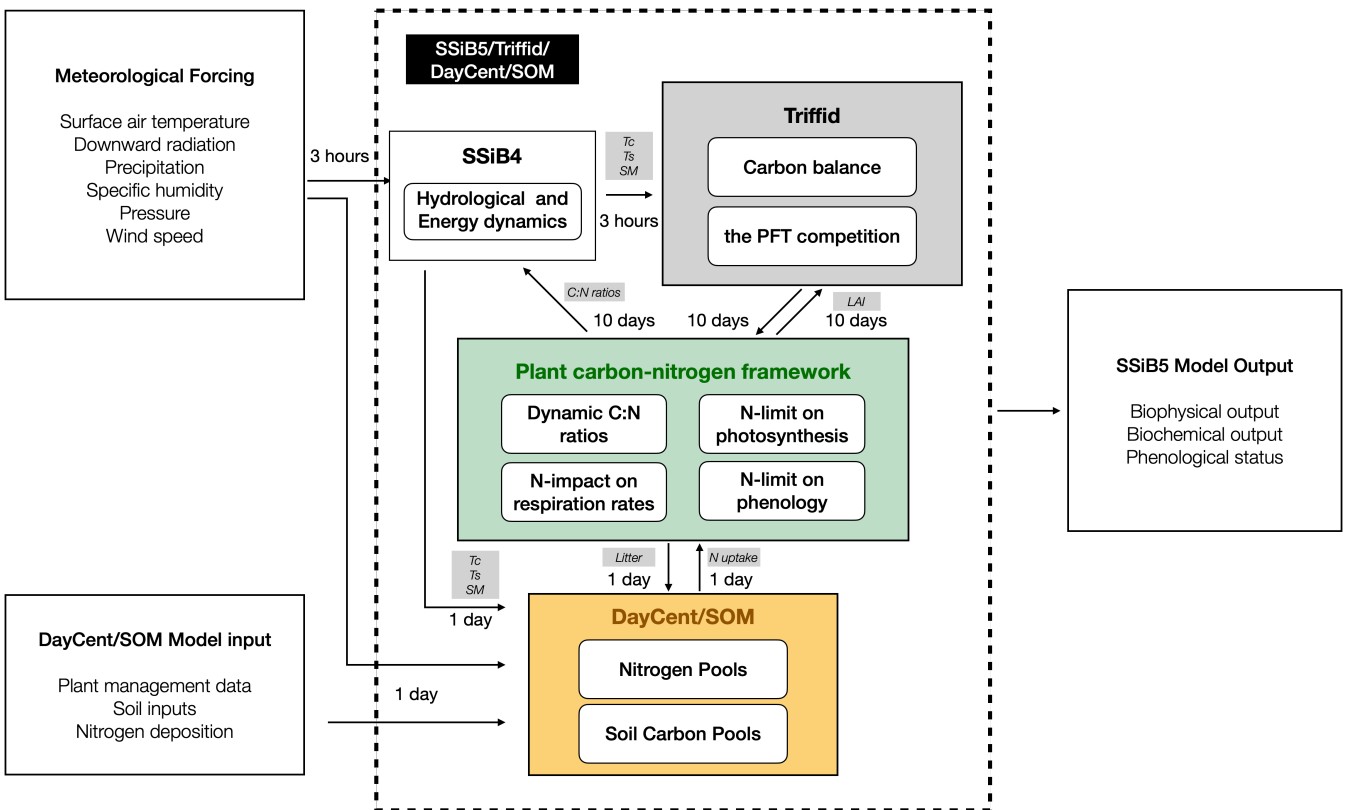

**Figure 3.** Flowchart of plant carbon-nitrogen interactions in SSiB5/TRIFFID/DayCent-SOM; main variables are listed between two modules. Notes: Tc: canopy temperature; Ts: land surface temperature; SM: soil moisture; GPP: gross primary productivity; Res: autotrophic respiration.

## 2.3 Model forcing and validation data

Long term measurements from flux tower sites with different PFTs and global satellite-derived products are employed as references to systematically assess the effects of this coupling framework and N limitation on the terrestrial carbon cycle. Flux tower sites data are presented in section 2.3.1. The global meteorological forcing and validation data are listed in sections 2.3.2 and 2.3.3 separately.

### 2.3.1 Ground measurement data

To validate the coupled model, twelve sites with representative biome types and climate zones were selected to evaluate the simulations of seasonal patterns of GPP, sensible heat flux, and latent heat flux. The driving data were a half-hourly dataset, including air temperature, specific humidity, wind velocity, air pressure, precipitation, and short- and long-wave radiation data



from the FLUXNET 2015 dataset (Pastorello et al., 2020). The geographical distribution of selected FLUXNET 2015 sites is displayed in Figure 4 and the detailed site information is listed in Table 3.

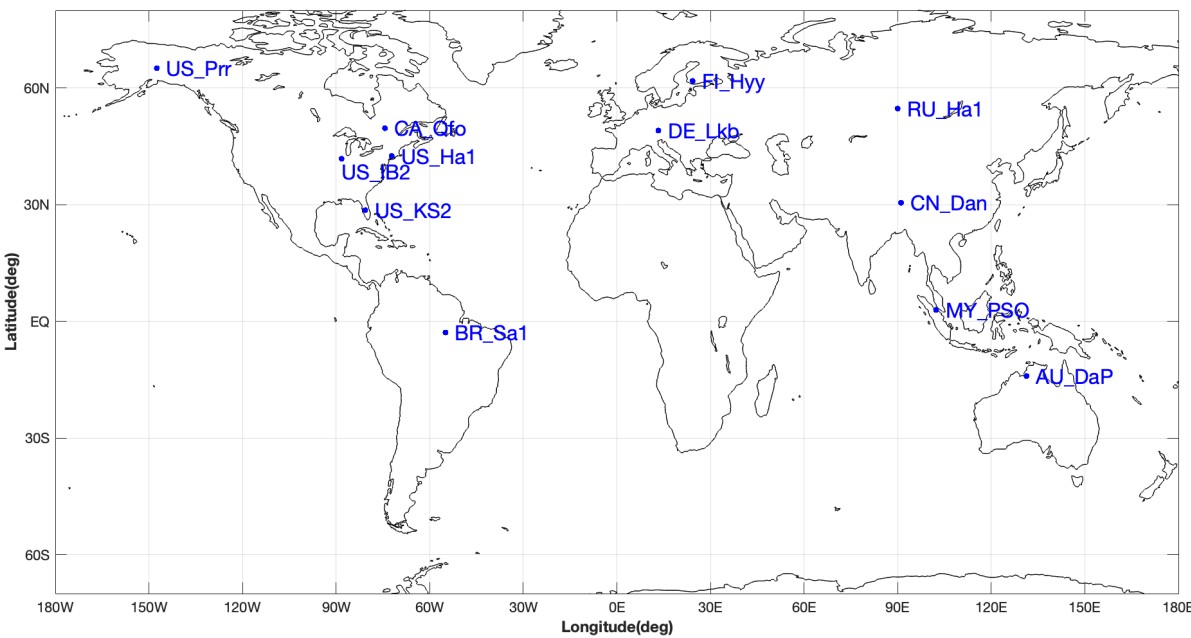


**Figure 4.** Geographical distribution of selected FLUXNET 2015 sites. The information of these FLUXNET sites is listed in Table 1.

**Table 3.** FLUXNET sites, latitude (LAT), longitude (LONG), plant function type (PFT), and time frame (Time) used for SSiB5/TRIFFID/DayCent-SOM model validation.

| Site_ID | Site name | LAT | LONG | PFT | Time |
|---|---|---|---|---|---|
| AU_DaP | Daly River Savanna | -14.06 | 131.32 | C4 grass | 2007-2013 |
| BR-Sa1 | Santarem-Km67-Primary Forest | -2.86 | -54.96 | Broadleaf Evergreen | 2002-2011 |
| CA_Qfo | Quebec - Eastern Boreal, Mature Black Spruce | 49.69 | -74.34 | Needleleaf Evergreen | 2003-2010 |
| CN-Dan | Dangxiong | 30.50 | 91.07 | C3 grass | 2004-2005 |
| DE_Lkb | Lackenberg | 49.10 | 13.30 | Needleleaf Evergreen | 2009-2013 |
| FI_Hyy | Hyytiala | 61.85 | 24.29 | Needleleaf Evergreen | 1996-2014 |
| MY_PSO | Pasoh Forest Reserve | 2.97 | 102.31 | Broadleaf Evergreen | 2003-2009 |
| RU_Ha1 | Hakasia steppe | 54.73 | 90.00 | C3 grass | 2002-2004 |
| US_Ha1 | Harvard Forest EMS Tower (HFR1) | 42.54 | -72.17 | Broadleaf deciduous | 1991-2012 |
| US_IB2 | Fermi National Accelerator Laboratory- Batavia (Prairie site) | 41.84 | -88.24 | C3 grass | 2004-2011 |
| US-KS2 | Kennedy Space Center (scrub oak) | 28.61 | -80.67 | Shrub | 2003-2006 |
| US_Prr | Poker Flat Research Range Black Spruce Forest | 65.12 | -147.49 | Needleleaf Evergreen | 2010-2014 |



### 2.3.2 Meteorological forcing data

The Princeton global meteorological dataset for land surface modelling (Sheffield et al., 2006) was used to drive SSiB4/TRIFFID global simulations from 1948 to 2007 at 1º x 1º spatial resolution and a 3-hour temporal interval. This dataset, including surface air temperature, pressure, specific humidity, wind speed, downward short-wave radiation flux, downward long-wave radiation flux, and precipitation, was constructed by combining a suite of global observation-based datasets with the National Center for Environmental Prediction/National Center for Atmospheric Research reanalysis data.

### 2.3.3 Global remote-sensing data

To assess the climatological status, variation, and trends of simulated LAI, two widely used global LAI products were used as references in this study: the Global Inventory Modeling and Mapping Studies (GIMMS) LAI and the Global LAnd Surface Satellite (GLASS) LAI. GIMMS-LAI is based on the third generation of Normalized Difference Vegetation Index (NDVI3g) from the GIMMS group and an Artificial Neural Network model (Zhu et al., 2013). GIMMS-LAI provides a 1/12-degree resolution, 15-day composites, and spans July 1981 to December 2011. GLASS-LAI is generated from Advanced Very High Resolution Radiometer (AVHRR) (from 1982 to 1999 with 0.05-degree resolution) and Moderate Resolution Imaging Spectroradiometer (MODIS, from 2000 to 2012 with 1 km resolution) reflectance data using general regression neural networks (Xiao et al., 2014). GIMMS and GLASS LAI and the meteorological forcing data for the overlap period of 1982 to 2007 were remapped to 1-degree spatial resolution and a monthly temporal interval.

The Model Tree Ensemble (MTE) GPP product (Jung et al., 2009) was used as a reference to evaluate simulated GPP. MTE is based on a machine learning technique in which the model is trained to predict the five C fluxes at FLUXNET sites driven by observed meteorological data, land cover data, and the remotely-sensed fraction of absorbed photosynthetic active radiation (Jung et al., 2009). The trained model was then applied at the grid scale driven by gridded forcing data. MTE-GPP data were resampled to 1-degree spatial and a monthly temporal resolution. However, the MTE data do not include $CO_2$ fertilization. Liu et al. (2019) discuss this issue and indicate that the lack of $CO_2$ fertilization mainly affects the trend. Since this paper focuses on climatological mean as well as differences between different experiments in which the $CO_2$ fertilization effect would be largely cancelled, the missing $CO_2$ fertilization in the FLUXNET-MTE is not a factor in interpreting our results.

## 3 Experimental design

To illustrate the reliability of the schemes which represent different processes of plant N in our framework, we first evaluated the model's short-term performance using in-situ measurements (section 3.2). Then, four sets of sensitivity experiments were designed to quantify the major effects of the plant N process and the relative contributions of different plant N processes on the terrestrial ecosystem carbon cycle (section 3.3).



### 3.1 Initial condition for the dynamic vegetation model

The initial condition of the dynamic vegetation SSiB4/TRIFFID needs to be obtained from a long-term equilibrium simulation (Zhang et al., 2015). There are different ways to initialize the surface condition for the quasi-equilibrium simulation. Following previous SSiB4/TRIFFID studies (Huang et al., 2020; Liu et al., 2019; Zhang et al., 2015), we set up the initial condition for the run using the SSiB vegetation map and SSiB vegetation table, which are based on ground surveys and satellite-derived information (Dorman and Sellers, 1989; Sellers et al., 1986; Xue et al., 2004; Zhang et al., 2015) with 100% occupation at

each grid point for the dominant PFT and zero for other PFTs. We then ran the SSiB4/TRIFFID model with the climate forcing and the atmospheric $CO_2$ concentration at 1948 level for 100 years to reach equilibrium conditions. The vegetation and soil conditions from the equilibrium results were used as the initial conditions for the subsequent model runs.

Determining the initial conditions for SSiB5/TRIFFIID/DayCent-SOM was carried out as described for SSiB4/TRIFFID with one additional step in order to initialize global soil C and N levels. We saved 60 years of daily litter C/N inputs and soil

temperature and moisture conditions from SSiB4/TRIFFID that were based on historical meteorological forcings (1948-2007). An offline version of DayCent-SOM was run for 2000 years for each grid cell using these 60 years of data, repeated over and over, to determine quasi-equilibrium soil C & N levels; these soil C and N values were read in by SSiB5/TRIFFIID/DayCent-SOM at the start of the global simulation in 1948.

### 3.2 Site-level validation

This paper focuses on the impact of N processes on the climatology of the global carbon cycle. Most current Dynamic Global Vegetation Models (DGVMs) are mainly focused on long-term (decadal to thousands of years or even longer) simulations at global scale; the diurnal and seasonal variations are not a subject for their modelling. Moreover, adequate long-term in-situ measurements are also not available for comparison. However, since the SSiB5/TRIFFID is a process-based model, we can evaluate the model's short-term performance using in-situ measurements.

Twelve sites with representative biome types and climates zones (Table 3 and Fig. 4) were selected to evaluate the simulations of seasonal patterns of fluxes over these sites. The site-level simulations were conducted by SSiB4/TRIFFID (a C-only model) and SSiB5/TRIFFID/DayCent-SOM separately to validate the model's performance. The model results were compared against observed daily data obtained by the flux tower including GPP, sensible heat flux, and latent heat flux.

### 3.3 Global 2-D offline control run and sensitivity runs

In this study, the SSiB4/TRIFFID and the SSiB5/TRIFFID/DayCent-SOM were applied to conduct a series of global 2-D offline runs (Table 4). All these runs employed the quasi-equilibrium simulation results as the initial condition, then were driven by the historical meteorological forcing from 1948 through 2007. The run using the SSiB4/TRIFFID is referred to as the control run (Exp. SSiB4 hereafter). Using the control simulation, we first evaluated the ability of the model to produce the climatology and variability of several biotic variables by comparing the results to multiple observation-based datasets. In



addition to the control run, six sets of sensitivity experiments were conducted to quantify the major effects of the N process and C-N interface coupling methodology on the C cycle. These sensitivity experiments were designed as follows:

(1) Nitrogen limitation on photosynthesis (Exp. NlPSN): The same meteorological forcing used for the control (Exp. SSiB4) drives the model, but dynamic C/N ratios and N limitation on $V_{c,max}$ (Eq. 6) are introduced. The difference between Exp. SSiB4 and Exp. NlPSN indicates the effect of N limitation on photosynthesis.

(2) Nitrogen impact on Respiration rate (Exp. NlResp): The model was driven by the same meteorological forcing used for Exp. SSiB4, but dynamic C/N ratio and N impacts on autotrophic respiration (Eq. 10) are introduced. The difference between Exp. SSiB4 and Exp. NlResp indicates the effect of N impact on respiration rate.

(3) Nitrogen limitation on Phenology (Exp. NlPhen): The model was driven by the same meteorological forcing used for Exp. SSiB4, but dynamic C/N ratio and N impacts on phenology (Eq. 13) are introduced. The difference between Exp. SSiB4 and
Exp. NlPhen indicates the effect of nitrogen limitation on phenology.

(4) SSiB5/TRIFFID/DayCent-SOM (Exp. SSiB5): The model was driven by the same meteorological forcing used for Exp. SSiB4, but all four C-N coupling processes in the framework, i.e., dynamic C/N ratio, N impacts on photosynthesis, autotrophic respiration, and phenology, are introduced. The difference between Exp. SSiB4 and Exp. SSiB5 indicates the effect of N dynamics, especially the sensitivity of C cycle variability and trend to N process coupling. Furthermore, the difference between
Exp.NlPSN and Exp. SSiB5 indicates the uncertainty (or possible errors) due to missing N effect on autotrophic respiration and phenology in the coupling framework.

Although the model runs were from 1948 to 2007, we only present the results from 1982-2007 to avoid spinning up for the SSiB5/TRIFFID/DayCent-SOM after SSiB4/TRIFFID and DayCent-SOM each reached their historical equilibrium conditions. Since the results from Exps. SSiB5 and NlPSN showed statistically significant differences from Exp. SSiB4 over many parts
of the world, in the following discussion we will mainly focus on these two experiments' differences with Exp. SSiB4.

**Table 4.** Experimental design

| 100-year equilibrium | *Initial condition* → | Real-forcing simulation 1948-2007 |
|---|---|---|
| *Fixed climatology forcing* | | *Transient forcing* |
| Control experiment | | **SSiB4:** Control experiment |
| | | **NlPSN:** Nitrogen limitation on photosynthesis(Vmax), Eq.6 |
| | | **NlResp:** Nitrogen impact on Respiration rate, Eq.10 |
| | | **NlPhen:** Nitrogen limitation on Phenology, Eq. 13 |
| | | **SSiB5:** including all four nitrogen processes |





## 4. Results

To test this framework, measurements from flux tower sites with different PFTs and global satellite-derived products from
1982-2007 are employed as references. The results from site simulation and global 2-D simulations are presented in sections
4.1 and 4.2., respectively. As mentioned in section 2, the framework takes some plant N metabolism processes into account.
To illustrate the relative contributions of different plant N processes on the terrestrial ecosystem carbon cycle, four sets of
sensitivity experiments were designed (Table 4). The analyses are presented in section 4.2.

### 4.1 Evaluations using the measurements from flux tower sites

Land models with dynamic vegetation and nitrogen processes normally focus on the long-term climate simulation with large
spatial  scales. .In this section, we validate the model performance for twelve sites with several years of simulation (Table 3)
to ensure that as a process-based model, the SSiB5/TRIFFID in the short term simulation is still able to properly represent the
surface processes at seasonal scales after introducing the DayCent-SOM through the interface coupling framework. This
evaluation also provides a glance at the model's performance at several sites with various climate and PFTs (Table 3) with
short-term data to gain preliminary confidence for further evaluation.

Figures 5, 6, and 7 show that both SSiB4 and SSiB5/TRIFFID/DayCent-SOM produce a reasonable seasonal cycle for GPP,
sensible heat, and latent heat fluxes, respectively, and that the results are close to observation. Table 5 summarizes the major
results. We use bias, root-mean-squared error (RMSE), as well as standard deviation to assess model performance against the
in-situ site measurements. When we evaluate the 12 sites average, the biases for GPP and sensible and latent heat fluxes are
decreased by about 7%, 18%, and 2 %, respectively. The average RMSEs over the 12 sites for these three variables are also
decreased by about 2%, consistent with the reduction in bias. Furthermore, the SSiB5/TRIFFID/DayCent-SOM produces
closer standard deviation for GPP, sensible heat flux, and latent heat flux than SSiB4/TRIFFID for the 12-site averages. By
and large, in these short-term simulations with specified initial vegetation conditions, both SSiB4 and SSiB5 produce
reasonable GPP and surface heat fluxes compared with in-situ measurements, but adding N processes (SSiB5) shows a slight
improvement for the 12-site average. Although these improvements are rather marginal (except the bias reduction for sensible
heat), the results nevertheless demonstrate that, with short-term simulation, model simulations' improvement is rather
consistent.

With closer checking of the SSiB4 to SSiB5 results at each site, the results display various characteristics. For instance, some
sites mainly show the improvement in both bias and RMSE, while others show improvement only in one or the other. Moreover,
while some sites show improvement in all three variables (GPP, latent and sensible heat fluxes), others only show improvement
for one or two variables. It should be pointed out that SSiB4 and SSiB5 are mainly used for global studies. For validation of
in-situ measurements, proper optimization of some site-specific soil and vegetation parameters is necessary (Xue et al., 1996,
1997). In this study, no model parameters were optimized during this validation exercise for a better fit between simulated
results and FLUXNET measurements. The discussions above lead us to conduct long-term experiments at a global scale to



comprehensively investigate the N process effect and to help understand mechanisms governing the global carbon cycle, which

will be discussed in the following section.





**Figure 5.** Simulated seasonal variations of GPP against observations at twelve FLUXNET sites representing different SSiB5 PFTs.

Note: the information about these FLUXNET sites is listed in Table 3.





**Figure 6.** Same as Figure 5, but for sensible heat flux.





**Figure 7.** Same as Figure 5, but for latent heat flux.






**Table 5.** The GPP, sensible heat flux, and latent heat flux intercomparisons of bias, standard deviation and RMSE between SSiB4 and SSiB5 over twelve sites.

| | Site_ID | Bias | | Standard deviation | | | RMSE | |
|---|---|---|---|---|---|---|---|---|
| | | SSiB4 | SSiB5 | Fluxnet | SSiB4 | SSiB5 | SSiB4 | SSiB5 |
| **GPP** | AU_DaP | 0.05 | -0.05 | 3.11 | 2.46 | 2.33 | 2.60 | 2.61 |
| **(g C d$^{-1}$)** | BR-Sa1 | -1.07 | -1.20 | 1.31 | 0.57 | 0.55 | 1.77 | 1.84 |
| | CA_Qfo | -0.05 | -0.11 | 1.71 | 1.99 | 1.92 | 0.78 | 0.75 |
| | CN-Dan | 0.70 | 0.08 | 0.92 | 1.08 | 1.03 | 0.80 | 0.33 |
| | DE_Lkb | 0.34 | 0.25 | 1.50 | 1.80 | 1.71 | 0.80 | 0.74 |
| | FI_Hyy | -0.11 | -0.22 | 2.93 | 3.47 | 3.32 | 1.51 | 1.44 |
| | MY_PSO | -1.02 | -1.20 | 0.65 | 1.28 | 1.21 | 1.63 | 1.72 |
| | RU_Ha1 | -0.24 | -0.27 | 1.29 | 1.31 | 1.27 | 0.69 | 0.69 |
| | US_Ha1 | 0.36 | 0.27 | 3.31 | 3.36 | 3.30 | 1.31 | 1.28 |
| | US_IB2 | 0.56 | 0.42 | 2.91 | 2.70 | 2.57 | 1.80 | 1.79 |
| | US-KS2 | -0.28 | -0.52 | 1.37 | 1.76 | 2.01 | 1.35 | 1.54 |
| | US_Prr | -0.08 | -0.10 | 1.43 | 1.30 | 1.28 | 0.86 | 0.86 |
| **12-site average** | | **0.41** | **0.38** | **1.87** | **1.92** | **1.88** | **1.33** | **1.30** |
| **Sensible** | AU_DaP | 32.47 | 23.13 | 28.26 | 19.64 | 21.05 | 36.24 | 36.32 |
| **Heat** | BR-Sa1 | 45.29 | 40.94 | 4.04 | 16.32 | 15.98 | 25.61 | 25.07 |
| **Flux** | CA_Qfo | -7.04 | -2.34 | 27.77 | 33.18 | 29.37 | 9.54 | 9.20 |
| **(W m$^{-2}$)** | CN-Dan | 17.96 | 18.53 | 14.44 | 22.38 | 20.75 | 25.60 | 26.99 |
| | DE_Lkb | -3.12 | 0.16 | 25.13 | 35.39 | 36.91 | 17.83 | 18.15 |
| | FI_Hyy | 5.53 | 7.20 | 28.17 | 33.57 | 33.63 | 8.99 | 10.91 |
| | MY_PSO | 20.49 | 10.86 | 10.03 | 11.30 | 11.98 | 39.22 | 37.99 |
| | RU_Ha1 | -0.14 | 0.84 | 21.71 | 39.19 | 38.02 | 29.42 | 29.67 |
| | US_Ha1 | -18.34 | -15.80 | 24.40 | 33.71 | 29.42 | 24.33 | 24.66 |
| | US_IB2 | 20.21 | 18.26 | 11.95 | 32.89 | 29.19 | 23.16 | 28.72 |
| | US-KS2 | 27.74 | 20.81 | 21.01 | 19.17 | 20.14 | 27.31 | 24.73 |
| | US_Prr | 8.10 | 9.35 | 20.93 | 36.84 | 35.45 | 12.02 | 12.01 |
| **12-site average** | | **17.20** | **14.02** | **19.82** | **27.80** | **26.82** | **23.27** | **22.70** |
| **Latent** | AU_DaP | -11.02 | -10.83 | 45.72 | 30.03 | 33.93 | 36.24 | 36.32 |
| **Heat** | BR-Sa1 | -20.47 | -19.82 | 16.15 | 9.44 | 8.47 | 25.61 | 25.07 |
| **Flux** | CA_Qfo | 2.21 | 0.96 | 18.06 | 18.63 | 17.56 | 9.54 | 9.20 |
| **(W m$^{-2}$)** | CN-Dan | -12.63 | -12.57 | 42.39 | 22.13 | 20.77 | 25.60 | 26.99 |
| | DE_Lkb | -7.39 | -10.00 | 22.81 | 24.57 | 20.79 | 17.83 | 18.15 |
| | FI_Hyy | -3.06 | -4.84 | 23.22 | 19.21 | 16.64 | 8.99 | 10.91 |
| | MY_PSO | -38.18 | -36.18 | 7.07 | 9.24 | 11.64 | 39.22 | 37.99 |
| | RU_Ha1 | -22.89 | -23.10 | 25.68 | 10.43 | 10.08 | 29.42 | 29.67 |
| | US_Ha1 | -11.94 | -13.14 | 27.06 | 15.53 | 14.71 | 24.33 | 24.66 |
| | US_IB2 | -12.90 | -17.38 | 36.91 | 24.68 | 20.70 | 23.16 | 28.72 |
| | US-KS2 | -17.74 | -13.41 | 27.63 | 20.28 | 19.65 | 27.31 | 24.73 |
| | US_Prr | -1.90 | -1.87 | 16.44 | 9.62 | 9.68 | 12.02 | 12.01 |
| **12-site average** | | **13.93** | **13.67** | **25.76** | **17.82** | **17.95** | **24.27** | **23.70** |



## 4.2 Evaluation of GPP and LAI at global scale

Since the SSiB model is mainly used for global study, in this section, we evaluate the model's performance using the observed global GPP and LAI data. The simulated GPP averaged over 1982-2007 is compared to FLUXNET-MTE GPP (Jung et al., 2011) to examine the impact of N processes and its coupling with C and ecosystem processes. Both SSiB4/TRIFFID (Exp. SSiB4) and SSiB5/TRIFFID/DayCent-SOM (Exp. SSiB5) capture the distribution of global GPP (Fig. 8) and its latitudinal distribution (Fig. 9a). The highest GPP occurs in the tropical evergreen forest and generally decreases with the increase in

latitudes in both the observations and the model simulation (Figs. 8 and 9a). Exp. SSiB4-simulated GPP has positive bias over many parts of the world (Fig. 8d), including tropical Africa and the North American and eastern Siberian boreal regions, but negative bias in some regions, mainly in the Amazon tropical forest. The simulated global GPP is 1082.36 g C $m^{-2}$ $yr^{-1}$ (Table 6), higher than the estimate, 862.86 g C $m^{-2}$ $yr^{-1}$ in FLUXNET-MTE (Jung et al., 2011). After introducing the N limitation for three processes, the SSiB5 reduced the positive bias in SSiB4 over many parts of the world (Figs. 8e, 8f, and 9a). Exp. SSiB5's

global GPP prediction, 941.81 g C $m^{-2}$ $yr^{-1}$, is closer to observations compared to Exp. SSiB4, with a 16.3% reduction in the bias (Table 6). Furthermore, the temporal correlation coefficients between observed and simulated monthly/annual mean GPPs are increased from 0.46/0.98 (Exp. SSiB4) to 0.50/0.99 (Exp. SSiB5), respectively (Fig. 10), showing improvement in simulation of the seasonal cycle in SSiB5. The correlation for interannual variability in SSiB4 is already very high (0.98). SSiB5 shows no substantial improvement.





**Figure 8.** The 1982-2007 average gross primary production comparison for (a) FLUXNET-MTE GPP (OBS), (b) SSiB4/TRIFFID (SSiB4), and (c) SSiB5/TRIFFID/DayCent/SOM (SSiB5), and difference between (d) SSiB4-OBS, and (e) SSiB5-OBS, (f) SSiB5-SSiB4.

Note: SCC indicates the spatial correlation coefficient between model simulation and satellite-derived datasets (OBS).





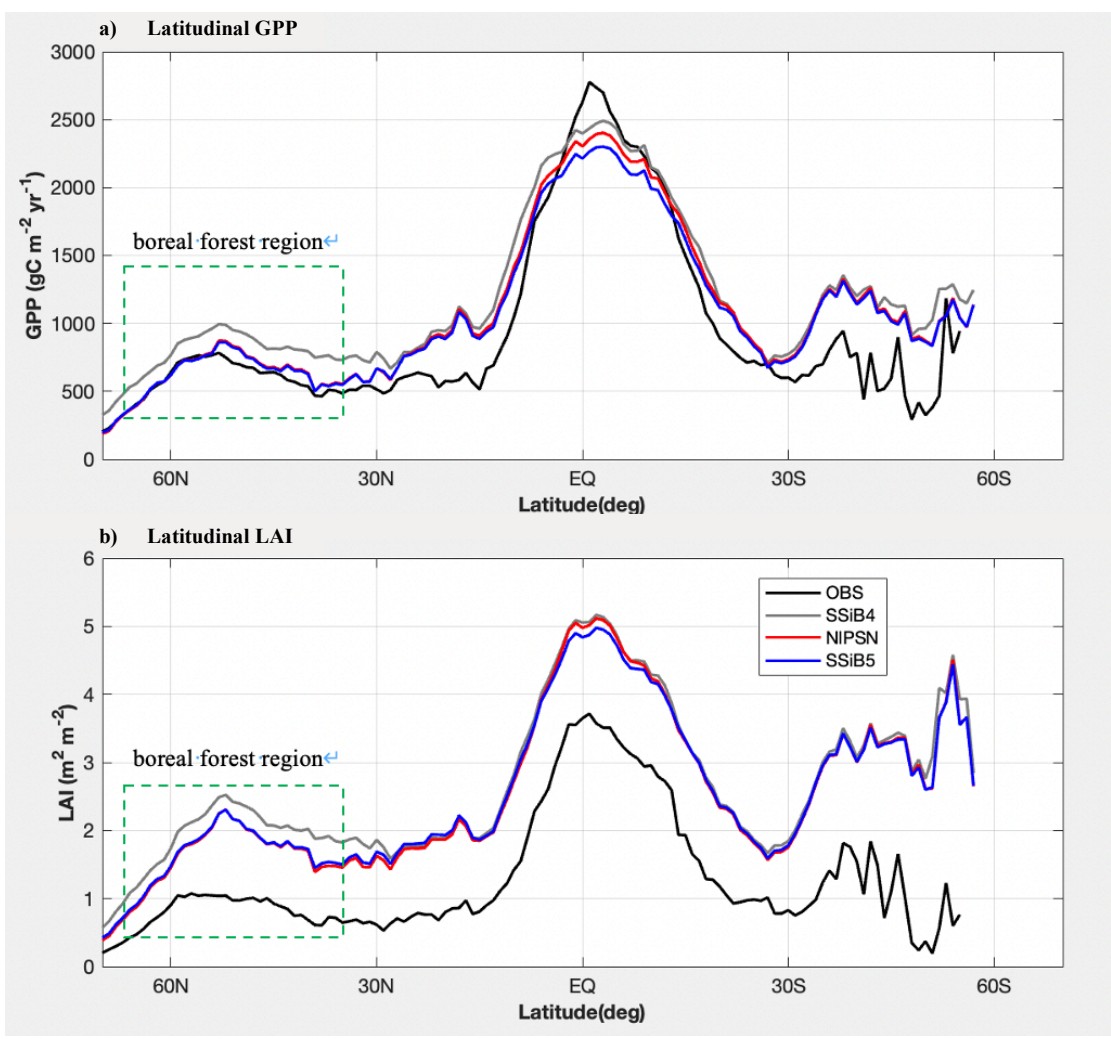

**Figure 9.** Intercomparisons of latitudinal LAI and GPP among OBS, SSiB4 (control), NlPSN (N limitation on photosynthesis only), and SSiB5 (all N processes) over the period 1982-2007.



**Table 6.** Regional and Global GPP for (a) FLUXNET-MTE GPP (observation), (b) SSiB4 (control), (c) NlPSN (N limitation on photosynthesis only) and (d) SSiB5 (N limitation on photosynthesis, autotrophic respiration, and phenology).

| Regions | Sub-regions | GPP Mean (gC m$^{-2}$ yr$^{-1}$) | | | | | | | |
|---|---|---|---|---|---|---|---|---|---|
| | | MTE | | SSiB4 | | NlPSN | | SSiB5 | |
| | | mean | bias | mean | bias | mean | bias | mean | bias |
| Arid and Semi-Arid Areas | West Africa | 893 | | 1147 | 254(28.5%) | 963 | 70(7.9%) | 915 | 22(2.5%) |
| | West NA | 438 | | 549 | 111(25.4%) | 454 | 16(3.5%) | 431 | -7(-1.6%) |
| | SA Savanna | 1665 | | 1860 | 195(11.7%) | 1763 | 98(5.9%) | 1675 | 10(0.6%) |
| | East Africa | 1228 | | 1533 | 306(24.9%) | 1427 | 199(16.2%) | 1356 | 128(10.4%) |
| | East Asian semi-arid | 1440 | | 1470 | 30(2.1%) | 1199 | -241(-16.7%) | 1139 | -301(-20.9%) |
| NH High-Mid Latitude Areas | NA High-Mid Latitude | 552 | | 814 | 262(47.6%) | 700 | 149(27.0%) | 665 | 114(20.6%) |
| | Eurasian High-Mid | 844 | | 966 | 122(14.5%) | 871 | 27(3.2%) | 827 | 16(-2.0%) |
| Equator | Amazon Basin | 2993 | | 2668 | -326(-10.9%) | 2631 | -362(-12.1%) | 2500 | -494(-16.5%) |
| | Southeast Asia | 2778 | | 2540 | -238(-8.6%) | 2419 | -359(-12.9%) | 2298 | -480(-17.3%) |
| | Equator Africa | 2522 | | 2645 | 123(4.9%) | 2611 | 89(3.5%) | 2481 | -42(-1.7%) |
| Subarctic Areas and Tibet | NA Subarctic | 234 | | 364 | 130(55.7%) | 240 | 6(2.4%) | 228 | -6(-2.7%) |
| | Eurasian Subarctic | 331 | | 484 | 153(46.2%) | 328 | -3(-1.0%) | 311 | -20(-6.0%) |
| | Tibet | 409 | | 561 | 153(37.3%) | 298 | -111(-27.2%) | 283 | 126(-30.8%) |
| *Global* | | **863** | | **1082** | **220(25.4%)** | **991** | **129(14.9%)** | **942** | **79(9.1%)** |

Note: the numbers in parentheses are relative biases: (bias/MTE mean)

The improvement, however, is not homogeneous over the globe but displays apparent regional characteristics. The GPP biases in tropical Africa, North American boreal region, South American savanna, and central U.S. show substantial reduction (Fig. 8f), which help improve the spatial distribution of SSiB5. The global spatial correlation coefficient increases from 0.88 to 0.90 (Fig. 8). Meanwhile, the GPP simulations in some regions, such as in the temperate East Asian mixed forest-grassland regions and in some areas in Siberia (Fig. 8), did not improve. In particular, the negative GPP bias in the Amazon is increased (Fig. 8f). This phenomenon has also appeared in the offline test in the Amazon site (the BR-Sa1 Site, Table 5). Du et al. (2020) indicate that phosphorus (P) has more effect in tropical areas. This paper mainly focuses on model development and preliminary global evaluation. More regional evaluation is necessary for further investigation, especially for the regions where the N limitation is not dominant.





**Figure 10.** Intercomparisons of global monthly/annual mean GPPs among OBS, SSiB4 (control), NlPSN (N limitation on photosynthesis only), and SSiB5 (all N processes) over the period 1982-2007.



**Figure 11.** Same as figure 10, but for LAI.

Furthermore, the N-limitation effect on the LAI simulation is also investigated. Both SSiB4 and SSiB5 produce reasonable spatial distribution compared with satellite-derived products (Figs. 12a-c). The highest LAI occurs in the tropical evergreen forest and decreases with latitude in both the observations and the model (Fig. 9b). Exp. SSiB5 also generally reduces the positive bias in simulated LAI compared to the control (Fig. 12f). The simulated LAI in Exp. SSiB4 has a global positive bias. After introducing three N limitation processes, the positive bias is reduced over most parts of the world (Fig. 12f). Globally, Exp. SSiB5 has an LAI bias of 0.94/1.12 for GIMMS/GLASS, respectively (Table 7), which is lower than the LAI bias of 1.26/1.44 for GIMMS/GLASS, respectively, in Exp. SSiB4, with a substantial 31.1% reduction in the bias (compared to GIMMS, Table 7). However, the positive bias still exists substantially over the globe (Fig. 12e). In a land model





intercomparison, the positive LAI prevailed in almost every dynamic vegetation model (Murray-Tortarolo et al., 2013). Our study shows that imposing N limitation is an adequate step to overcome the dynamic vegetation models' systematic LAI positive bias, but the issue is still not solved and requires more investigation. In addition, the correlation coefficients between observed and simulated monthly/annual average LAIs are improved from 0.49/0.97 (Exp. SSiB4) to 0.51/0.98 (Exp.SsiB5)
(Fig. 11).

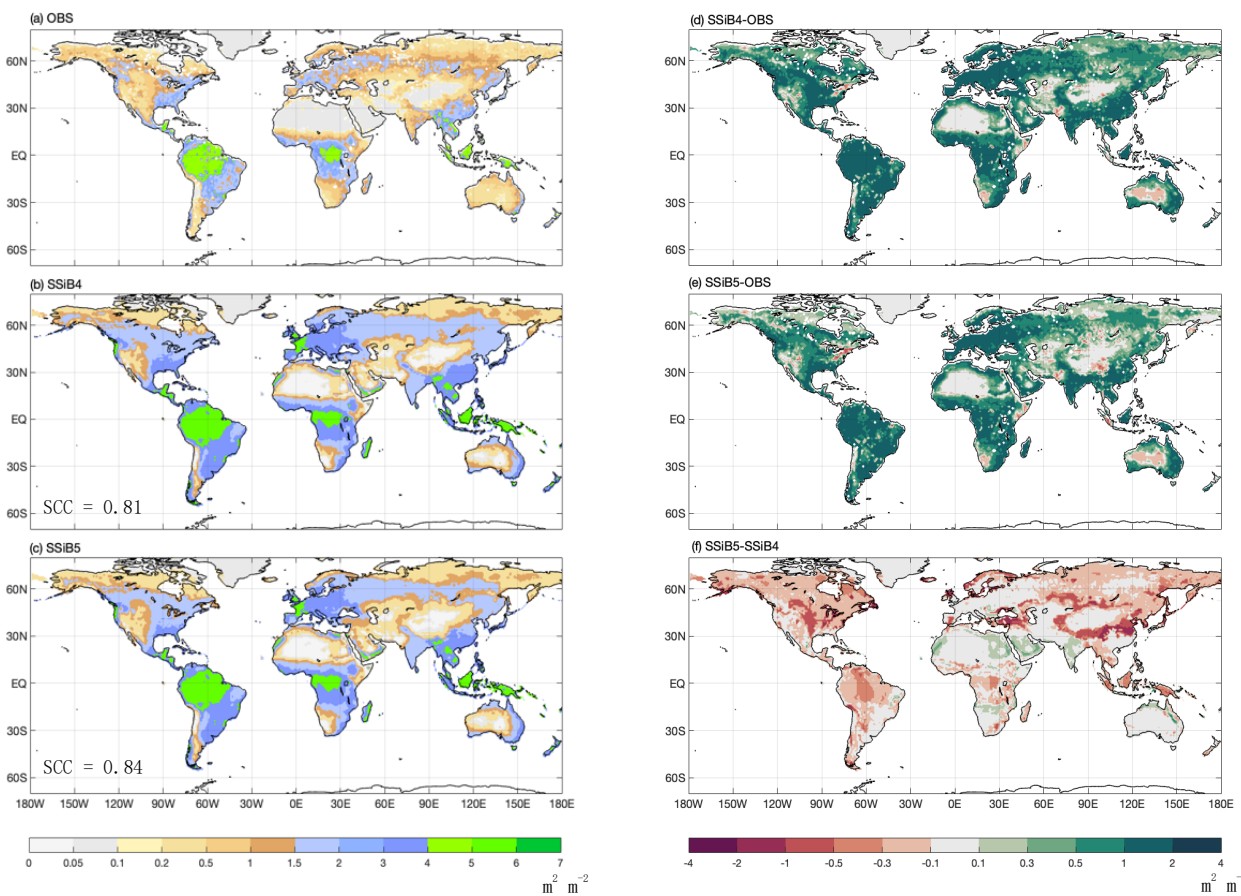

**Figure 12.** Same as Figure 8, but for LAI.

Note: SCC indicates the spatial correlation coefficient between model simulation and GIMMS LAI (OBS).


It is interesting to note that despite the global general LAI reduction, the SsiB5 slightly increased LAI estimation in North Africa and India (Fig. 12). The N impacts on phenology and respiration cause a slight shift in the vegetation from shrub (N. Africa) or C4 grass (India) to C3 grass in these areas, which contributes to the GPP and LAI increase (Fig. 13). Furthermore, in areas such as the Amazon, East Asian mixed forest-grassland regions, , SSiB5 only improves the LAI simulation, not the



GPP simulation. We will further identify the effect of N limitation on the photosynthesis process and other processes on simulated GPP and LAI. More observational data are necessary to gain more understanding.

**Table 7.** Regional and Global LAI for (a) GIMMS LAI (observation), (b) GLASS LAI (second observation), (c) SSiB4 (control), (d) NlPSN (N limitation on photosynthesis only) and (e) SSiB5 (N limitation on photosynthesis, autotrophic respiration, and phenology). The bias is
relative to GIMMS LAI.

| Regions | Sub-regions | LAI Mean ($m^2\,m^{-2}$) | | | | | | | | | |
|---|---|---|---|---|---|---|---|---|---|---|---|
| | | GIMMS | | GLASS | | SSiB4 | | NIPSN | | SSiB5 | |
| | | mean | bias | mean | bias | mean | bias | mean | bias | mean | bias |
| Arid and Semi-Arid Areas | West Africa | 1.08 | | 1.01 | -0.07(-6.5%) | 2.04 | 0.96(88.9%) | 1.89 | 0.81(75.0%) | 1.73 | 0.65(60.2%) |
| | West NA | 0.62 | | 0.49 | -0.13(-21.0%) | 1.38 | 0.76(122.6%) | 1.18 | 0.56(90.3%) | 1.09 | 0.47(75.8%) |
| | SA Savanna | 1.99 | | 1.91 | -0.18(-4.0%) | 3.34 | 1.35(67.8%) | 3.23 | 1.24(62.3%) | 2.97 | 0.98(49.2%) |
| | East Africa | 1.59 | | 1.55 | -0.04(-2.5%) | 3.02 | 1.43(89.9%) | 2.89 | 1.30(81.8%) | 2.66 | 1.07(67.3%) |
| | East Asian semi-arid | 1.60 | | 1.36 | -0.24(-15.0%) | 3.35 | 1.75(109.4%) | 2.84 | 1.24(77.5%) | 2.61 | 1.01(63.1%) |
| NH High-Mid Latitude Areas | NA High-Mid Latitude | 0.84 | | 0.49 | -0.35(-41.7%) | 1.91 | 1.07(127.4%) | 1.66 | 0.82(97.6%) | 1.53 | 0.69(82.1%) |
| | Eurasian High-Mid | 1.14 | | 0.57 | -0.57(-50.0%) | 2.29 | 1.15(100.9%) | 2.08 | 0.94(82.5%) | 1.91 | 0.77(67.5%) |
| Equator | Amazon Basin | 4.19 | | 4.08 | -0.11(-2.6%) | 6.01 | 1.82(43.4%) | 5.98 | 1.79(42.7%) | 5.50 | 1.31(31.3%) |
| | Southeast Asia | 3.93 | | 3.88 | -0.05(-1.3%) | 4.68 | 0.75(19.1%) | 4.68 | 0.75(19.1%) | 4.31 | 0.38(9.7%) |
| | Equator Africa | 3.83 | | 3.76 | -0.07(-1.8%) | 5.74 | 1.91(49.9%) | 5.72 | 1.89(49.3%) | 5.27 | 1.44(37.6%) |
| Subarctic Areas and Tibet | NA Subarctic | 0.32 | | 0.14 | -0.18(-56.3%) | 0.71 | 0.39(121.9%) | 0.51 | 0.19(59.4%) | 0.47 | 0.15(46.9%) |
| | Eurasian Subarctic | 0.33 | | 0.12 | -0.21(-63.6%) | 0.87 | 0.54(163.6%) | 0.65 | 0.32(97.0%) | 0.60 | 0.27(81.8%) |
| | Tibet | 0.64 | | 0.54 | -0.10(-15.6%) | 1.36 | 0.72(112.5%) | 0.81 | 0.17(26.6%) | 0.75 | 0.11(17.2%) |
| *Global* | | 1.18 | | 1.00 | -0.18(-15.3%) | 2.44 | 1.26(110.8%) | 2.31 | 1.13(95.8%) | 2.12 | 0.94(79.7%) |

Note: the numbers in parentheses are relative biases



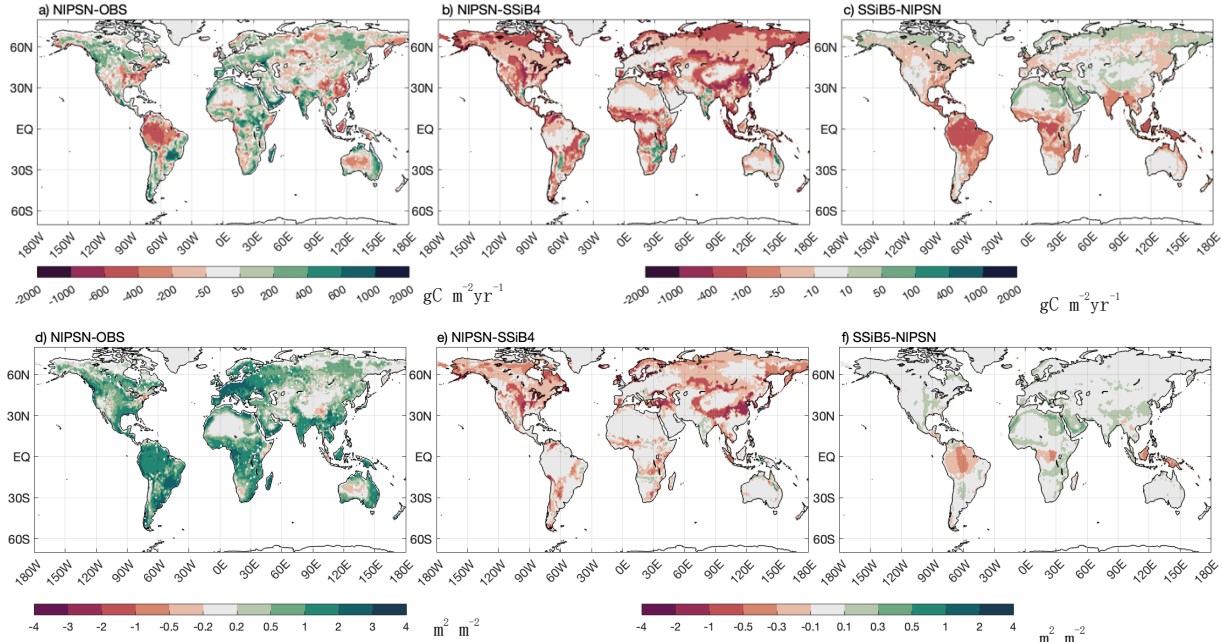

**Figure 13.** The 1982-2007 average gross primary production difference (a) NIPSN-OBS, (b) NIPSN-SSiB4, (c) SSiB5-NIPSN, and leaf area index difference (d) NIPSN- OBS, (e) NIPSN- SSiB4, (f) SSiB5- NIPSN

Note: NIPSN is N limitation on photosynthesis (Vc, max) only.

We imposed N limitation on several processes. Among them, Exp. NIPSN shows the largest impact. The results from Exp. NIPSN, which apply Eq. (6) to scale down the $V_{c,max}$, are discussed here. Please note, the differences between Exp. SSiB5 and Exp. NIPSN show the effects due to another two N limitation processes (Eqs. 10 and 13). Exp. NIPSN has a lower global GPP bias (128.52 g C m$^{-2}$ yr$^{-1}$) compared to FLUXNET-MTE estimates than Exp. SSiB4 does (219.50 g C m$^{-2}$ yr$^{-1}$) (Fig. 13, Table 6), but it is larger than Exp. SSiB5, in which the bias is 79 g C m$^{-2}$ yr$^{-1}$ (Table 6). In addition, Exp. NIPSN has a global LAI

bias of 1.13 (Fig. 13, Table 7), which is also lower than the LAI bias in Exp. SSiB4 (1.26), but higher than Exp. SSiB5 (0.94). The largest reductions in the magnitude of the LAI bias are in North America, Eurasian continents, and tropical Savanna regions in South America and Africa (Figs. 13b and 13e). That said, the N limitation on photosynthesis plays a dominant role, contributing to about 65%/41% of the improvement for the GPP/LAI simulations in Exp. SSiB5, respectively. Adjusting $V_{c,max}$ is the most direct and process-based approach based on physiology and yields the largest impact. But N limitation on the other

two processes is still substantial. The N limitations on respiration and phenology have the most impact in tropical forest and savanna regions (Figs. 13c and 13f). For GPP, they also reduce the positive bias over boreal regions and the negative bias in the polar regions.



## 5 Discussion and Conclusions

This study presents improvements in modelling the C cycle by introducing plant N processes into the SSiB5/TRIFFID/DayCent-SOM, using DayCent-SOM to obtain the amount of N available to plants and plant soil N uptake. The approach presented in this study can also be applied to other models with similar physical and biological principles. The new C-N coupling framework allows us to use dynamic C/N ratios to represent plant resistance and self-adjustment, which allow adaptations in the stoichiometry of C and N. Since these processes can increase nutrient use efficiency and reduce the

impact of N limitation through N remobilization and resorption, N-limit effect would not linearly nor instantaneously respond to available N content. A linear relationship between N limitation factor and available N is valid only when N availability is not sufficient for the minimum N demand for new growth. This is an advantage of our approach. That said, with the new model structure, N impacts on GPP are predicted directly but not linearly with leaf N content, which is affected by the state of plant growth, autotrophic respiration, and plant phenology.

By comparing site-level results from SSiB4 and SSiB5 to FLUXNET GPP and surface heat fluxes from twelve sites with representative biome types and climate zones, we gained confidence in the ability of the new N processes to enhance global model performance. We also evaluated the model performance against global satellite product data sets for GPP and LAI. In general, with the new plant C-N coupling framework, SSiB5/TRIFFID/DayCent-SOM produces significantly less absolute bias for GPP and LAI than the baseline version of SSiB4/TRIFFID (without N processes), a global decrease of the bias in GPP

and LAI by 16.3% and 27.1%, respectively. The main improvements are found in tropical Africa and the boreal forest. The more realistic representation of dynamic C/N ratios and plant C-N framework leads to general improvements in SSiB5/TRIFFID/DayCent-SOM's global C cycling simulations. From the perspective of plant physiology (Högberg et al., 2017), the downregulation of the canopy photosynthetic rate based on the available mineral N for growth of plant tissues is more reasonable than the simple and direct downregulation of GPP or NPP. In fact, we have conducted a test to directly

downscale GPP and NPP, and our simulation results (not shown) support this viewpoint. This coupled model can better reproduce observed state variables and their emergent properties (such as GPP, NPP, LAI, and respiration). Despite the general improvement globally, the GPP simulation in the temperate East Asian mixed forest-grassland regions seems to get worse with SSiB5 compared to SSiB4. In some regions, such as the Amazon, while the SSiB4 produced lower GPP than observations, the imposed N limitation in SSiB5 would further increase the bias in the regions. This mismatch is a common issue reported in a

number of publications (Anav et al., 2015; Liu et al., 2019; Piao et al., 2013). Further investigations are necessary.

Recently, the important influence of phosphorus availability on the terrestrial ecosystem carbon uptake has been increasingly realized. The recently initiated ecosystem-scale manipulation experiments in phosphorus-poor environments (Fleischer et al., 2019) call for the need for new phosphorus enabled LSMs to keep track of these actions (Goll et al., 2017; Reed et al., 2015). We plan to incorporate other plant processes, such as plant/soil phosphorus processes, to further improve performance of the

model in the future.



*Data availability.* The evaluation/reference data sets from model data discussed in this paper are archived at
https://doi.org/10.5281/zenodo.7196869


*Code availability.* The source code of biophysical-ecosystem-biogeochemical model, SSiB version5/TRIFFID/DayCent-SOM
is archived at https://doi.org/10.5281/zenodo.7297108

*Author contributions.* ZX, YX, MH, and YL designed the coupling strategy between SSiB4/TRIFFID and DayCent-SOM. ZX

conducted the simulation with suggestions from YX, WG, and WP. ZX, YX, and MH drafted the text and ZX made the figures.
All authors (ZX, YX, WG, MH, YL and WP) have contributed to the analysis and the text.

C*ompeting interests.* The authors declare that they have no conflict of interest.

*Acknowledgments* This study is supported by the National Science Foundation, Division of Atmospheric and Geospace
Sciences (Grant No. AGS-1419526, AGS-1849654), and the Fundamental Research Funds for the Central Universities (Grant
No. 14380172). The authors acknowledge the use of the Cheyenne supercomputer (https://doi.org/10.5065/D6RX99HX,
Computational and Information Systems Laboratory, 2019), provided by NCAR CISL, for providing HPC resources.



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
