# Peer review of "Development of a plant carbon-nitrogen interface coupling framework in a coupled biophysical-ecosystem-biogeochemical model (SSiB5/Triffid/DayCent-SOM v1.0)"

_EGUsphere, 2023_

## Author Comment (AC1)

**Response to Anonymous Referee #1**

**General comments:**

This study addresses the critical need to improve the representation of nitrogen cycle in land biogeochemical processes which is a crucial aspect of modelling development. The manuscript presents a novel approach by integrating a soil organic matter and nutrient cycling model to advance simulating coupled global carbon-nitrogen cycles in a process-based dynamic vegetation model. The authors demonstrate that the coupled model version shows in general improvement of GPP, LAI, and heat fluxes validated against site-level and global observations, compared to the previous model version. It is certainly a timely work with supporting analysis, although only partially reflecting the performance of the revised model. There are several concerns that the authors should address to enhance the manuscript.

Reply: Thank you very much for your comprehensive and constructive reviews. We appreciate your effort and acknowledge your review in the paper's "Acknowledgment".

**Main concerns:**

1. The framework implemented in the study revolves around processes in terrestrial N cycles, more specifically about plant N demand and stress. However, the relevant processes are overly simplified when describing the necessity to modify current representations in the models.

Reply: Thank you for the comments and suggestions. In Main Concern 5, the reviewer also complains of a related issue: "The manuscript contains erroneous, redundant, or repetitive expressions throughout. Especially in the method section, there is a substantial amount of text either already addressed in the introduction or better suited for the discussion". Since these two issues both contribute to the presentation problem of model parametrization, we address these issues together.

To more comprehensively describe the necessity of modifying the current relevant parameterizations, we have added more relevant information to Section 2.2.1-2.2.5. Moreover, because our discussions on the methodology in the previous version spread across several places in the paper, which causes repetition and redundancy but is not comprehensive in either place, we reorganize the introduction and Section 2.2.1-2.2.5 to arrange the discussion on these processes more concentrating on relevant sections.

In this way, we hope that the description for each process is much clearer and more comprehensive and eliminates some redundancies and repetitions.

In the revised introduction, we mainly emphasize the necessity of including proper N processes in the Earth System Model (ESM) in general and move more detailed discussions on relevant processes (parameterizations) to the related sections 2.2.1-2.2.3. Moreover, we have attempted to improve the accuracy of the description of the development history of N process modeling and reduce the number of citations of relevant papers, as suggested by the review.

The following are a few examples in our revisions.

In the introduction, we revised the N-process model development as follows:

*New Lines 54-69: "Adequate C-N coupling in plant N processes has been indicated as an area that still needs intensive investigation (Thum et al., 2019; Ghimire et al., 2016; Goll et al., 2017; Yu et al., 2020; Zaehle et al., 2015; Zhu et al., 2019). The fundamental aspects of N cycling for terrestrial biosphere models, such as N limitation of vegetation growth, strategies in which vegetation invests C to increase the N supply under N-limited conditions, and N limitation of decomposition, have been identified as important challenges for representing N cycling in terrestrial biosphere models (Meyerholt et al., 2020; Peng et al., 2020; Zaehle et al., 2015). Some key plant N processes, such as N limitation on GPP, the effect of biomass N content on autotrophic respiration, plant N uptake, ecosystem N loss, and biological N fixation, have been introduced into LSMs with various complexities to determine the effects of N limitation in current land models. These methods include, for instance, using N to scale down the photosynthesis parameter V(c, max) (Ghimire et al., 2016; Zaehle et al., 2015) or potential GPP to reflect N availability (Gerber et al., 2010; Oleson et al., 2013; Wang et al., 2010), defining the C cost of N uptake (Fisher et al., 2010) and optimizing N allocation for leaf processes (Ali et al., 2015). The wide variety of assumptions and formulations of N cycling processes and C-N coupling reflects knowledge gaps and divergent theories, and further investigation is imperative (Kou-Giesbrecht, S., et al. 2023). The coupling of N processes is still an area of model development. In the latest Coupled Model Intercomparison Project Phase 6 (CMIP6, Eyring et al., 2016), although there were 112 different coupled models with various land surface models from 33 research teams, only about 10 models incorporated an N cycle module (Arora et al., 2020)."*

In Section 2.2.2 "Dynamic C/N ratio", some relevant information in the Introduction has been moved to this section. We first briefly describe the C/N ratio in the natural

world and the current status of C/N ratio modeling with proper citations. We then added more information for our modeling.

In Section 2.2.3 "Effects of N limitation on photosynthesis", we also moved the discussion of the parameterization of this issue from the Introduction to this section and made many revisions to improve our description of our model.

In Section 2.2.4 "Improvement of nitrogen impact on respiration rates", we added more information to support our approach and how our parameterization was obtained.

In Section 2.2.5 "N limitation on LAI based on plant phenology", we added more information on why this is an important issue, which was largely missing in the previous version.

We hope that with these major revisions, the issues that concern the reviewer will be properly addressed.

2.    Although it is important to evaluate the impacts on C and heat fluxes, more information on how the dynamic representation of the C/N ratios alter the N cycles would be very pertinent and interesting to report, provided the model outputs include relevant variables to describe the processes. Otherwise, there would remain a logic gap to make sense of the differences in C and heat variables between the new and old model versions.

Reply: We understand the reviewer's idea that it is interesting to compare the fixed C/N ratio and dynamic C/N ratio effects. In fact, this was also our original consideration. However, it is difficult to design such comparisons. Our parameterizations of the effects of N limitation on three processes (photosynthesis, phenology, and respiration) (Section 2.2.3-2.2.5) are based on the dynamic C/N ratio. For a fixed C/N ratio, some of these have to be changed. As such, it is difficult to just specify a fixed C/N ratio but with other parameterizations still being associated with a dynamic C/N ratio. Simply specifying a fixed C/N ratio may mislead readers.

In the revised paper, at the beginning of Section 2.2.2, we present many studies demonstrating that the dynamic C/N ratio is a phenomenon that exists in real biogeochemical processes, which we hope can provide a background and justification for the dynamic C/N ratio approach.

3.  The need to evaluate plant C processes under the modified N processes is well motivated in the introduction. However, the connection between N processes and heat fluxes is absent.

Reply: The reviewer raises a very important issue. As an NSF Climate Dynamics Program-supported project, how the N process influences the water and energy cycle is an important subject. In fact, we have presented the latent heat flux and sensible heat flux with/without N limitation in Table 6 and Figures 6 and 7. For the 13-site average, the results with N limitation showed only slight improvement.

This is because there are three components in our model that contribute to the total latent heat flux (as shown in Fig. 7 and Table 6). They include transpiration from the canopy, direct evaporation from the leaf due to interception loss of precipitation, and soil evaporation. In the offline test, the atmospheric demand is fixed; when transpiration is reduced/increased due to the change in photosynthesis process (caused by the N limitation), soil evaporation must change to satisfy the atmospheric demand. This change is not linear because the sensible heat flux also changes. As such, with fixed demand and radiative forcing from the atmosphere, it is difficult to properly assess the effect on heat flux. At the end of this paper, we added the following lines:

*New line 633-636. "Finally, this is an offline experiment in which the atmospheric forcing (such as downward radiation) is fixed. With a fixed atmospheric demand, the heat flux response due to the N limitation effect is also limited, as shown in section 4.1. A comprehensive assessment of the effect of N limitation on heat fluxes and atmospheric circulation needs to be conducted in a fully coupled atmosphere–land model."*

4.  The introduction made a leap from "C-only models and dynamic vegetation models generally miss the inclusion of N processes" to "this new framework not only consider N processes but also has a more realistic way to represent the processes with dynamic C/N ratios". Often dynamic vegetations include N cycles since decades, see e.g., Kou-Giesbrecht, S., et al. (2023) 10.5194/esd-14-767-2023. The current state of modelling C-N cycles is therefore misrepresented and the progress in other models is under recognized, although cited Davies-Barnard et al 2020 the authors themselves.

Reply: Thank you for pointing this out. We have revised this part of the introduction. Please see our response to Main Concerns #1.

5.  The manuscript contains erroneous, redundant, or repetitive expressions throughout. Especially in the method section, there is a substantial amount of text either already addressed in the introduction or better suited for the discussion.

Reply: We apologize for these writing issues. We have revised and reorganized the paper, especially the introduction and methodology sections (2.2.1-2.2.5), to make the statement accurate and avoid redundancies. Please see our response to Main concern #1 for more information.

**Minor points:**

1.   Suggest revising the title to make it more concise.

Reply: Done. We have revised this title to *"Development of a plant carbon-nitrogen interface coupling framework in a coupled biophysical-ecosystem-biogeochemical model (SSiB5/Triffid/DayCent-SOM v1.0")* and eliminated *"Its parameterization, implementation, and evaluation"*.

2.   The manuscript may benefit from additional analysis relating to the improved representation on N limitation for different PFTs.

Reply: We agree with the reviewer that it is a very good idea to include the analysis relating to the improved representation of N limitation for different PFTs. In fact, this information was included in our original manuscript. However, based on the editor's instructions for this paper's resubmission, the current paper is mainly focused on describing model development. The discussion on the scientific issues in the previous submission has been removed per the editor's suggestion.

3.   The use of "wood" or "stems/wood" as a plant organ can be misleading. In Table 1 they are then listed as "component" where it is also false to list "wood" under grasses. Please be consistent with the common terms and stick with stem.

Reply: Done. We now use "stem" throughout the paper.

4.   Suggest adding some information on tundra shrub as it is not covered in the validation sites (Table 3).

Reply: Thank you for the suggestion. Our validation sites were limited to the AmeriFlux sites. We now include a tundra site (Lund *et al*., 2012) for validation. The figure attached below is also included in the revised Section 4.1. The results are consistent with those at other sites and are shown in the revised Table 6 (previously Table 5).

**Table R1.** Tundra site information used for model validation.

| Site name | LAT | LONG | PFT | Time |
|---|---|---|---|---|
| Zackenberg Heath | 74.47 | -20.55 | tundra | 2000-2014 |

[Figure]

**Figure R1**. Simulated seasonal variations in GPP, sensible heat, and latent heat against observations at the tundra site.

**References:**

Lund, M., Falk, J. M., Friborg, T., Mbufong, H. N., Sigsgaard, C., Soegaard, H. and Tamstorf, M. P.: Trends in CO2 exchange in a high Arctic tundra heath, 2000–2010, J. Geophys. Res., 117(G2), G02001, 2012.

5.  The figure qualities are not consistent.

Reply: To improve the figure quality, we utilized MATLAB to redraw all the figures, employed the same image resolution parameters for the output, and used the PNG format instead of the PDF format for storage.

6.  When dealing with a variable having the unit of per area (e.g., Navail, g N m-2), the soil depth is essential information, however not clearly indicated in the manuscript.

Reply: Thank you for your careful review. We have added this information to the revised text as follows:

*New Lines 227-228:* *"The DayCent-SOM only provides the total available nitrogen ($N_{avail}$)* for the plant within one grid box (the soil is 3.2 m in depth), *which consists of several PFTs."*

7. The authors are strongly suggested to select references carefully instead of piling them up excessively, such as with the 17 citations in L54 to 57 and 12 citations in L40.

Reply: We apologize for this. We have selected references more carefully and deleted some as suggested, as shown in the example below.

*New Lines 50-53:* *"As such, the N cycle and its effect on C uptake in the terrestrial biosphere have been incorporated into land surface models (LSMs) of ESMs (Davies-Barnard et al., 2020; Kou-Giesbrecht et al, 2023) with various representations of N processes (Ali et al., 2015; Asaadi et al., 2021;* *; Clark et al., 2011; Davies-Barnard et al., 2020; Ghimire et al., 2016; Goll et al., 2017;*  *Lawrence et al., 2019;*  *Oleson et al., 2013; Smith et al., 2014; Thum et al., 2019;*  *Wiltshire et al., 2020;* *)."*

*New Lines 35-39:* *"To study these processes, the land surface components of Earth System Models (ESMs) have evolved from those that represent only physical processes (i.e., hydrology and the energy cycle) to those that include the terrestrial carbon I cycle, vegetation dynamics, and nutrient processes(Cox, 2001; Dan et al., 2020; Foley et al., 1998;*  *Oleson et al., 2013;*  *Sellers et al., 1996; Sitch et al., 2003; Wang et al., 2010; Zhan et al., 2003 ). "*

8. Please clarify several terms in the paragraph of L70-87, including "plant resistance on photosynthesis …" (as in it does not make sense to call it resistance on photosynthesis but more like resistance on the reduction of photosynthesis capacity or potential photosynthesis rate, not to be confused with photosynthesis rate, under N limitation), "C/N interactions" (as in if it is about the C to N ratio and something else, or the interactions between some C processes and N processes), "self-adjustment" (as in how such behaviors differ from being simply considered as "responses"), and "fertility" (as in if it refers to soil fertility or plant fertility which is not a well-known term).

Reply: Thank you for your careful review. In the revised paper,

"Plant resistance on photosynthesis" has been replaced with more proper words/presentations.

"C/N interaction" has been revised to "C and N process interaction".

"Self-adjustment" is replaced by "response", "adaptation", or eliminated.

"Fertility" was changed to "plant fertility".

**Line-specific comments:**

1. L38, suggest changing to only "physical processes" instead of "biophysical processes" as the commonly simplified land representation in ESMs does not include biological processes as the authors listed themselves.

Reply: Done.

*New Line 36:* *"To study these processes, the land surface components of Earth System Models (ESMs) have evolved from those that represent only physical processes (i.e., hydrology and the energy cycle) to those that include the terrestrial carbon (C) cycle, vegetation dynamics, and nutrient processes (Cox, 2001; Dan et al., 2020; Foley et al., 1998; Oleson et al., 2013; Sellers et al., 1986; Sitch et al., 2003; Wang et al., 2010). "*

2. L47, suggest changing "Those C-only models" to "The C-only models" as in not all those models mentioned prior, i.e., land process models and dynamic vegetation models, are C-only.

Reply: Done.

*New Line 45:* *"The C-only models ignore significant nitrogen (N) impacts and therefore overestimate C sequestration by terrestrial ecosystems under climate change (Peñuelas et al., 2013; Zaehle et al., 2015)".*

3. L72 "dynamic plant C/N ratio" is not necessarily a concept. According to the authors, it should be a more realistic representation than fix ratios.

Reply: Done.

*New Line 73:* *"The dynamic plant CNR is a more realistic representation than the fixed plant CNR in assessing the effect of N limitation on plant C processes and interactions between plant C and N processes."*

4. L73-74, please revise these two sentences. Suggest removing "Due to their relative immobility". Suggest changing "A deficiency of any type of nutrient" to "Nutrient deficiency".

Reply: The sentence has been revised and moved to Section 2.2.2.

*New Line 174* *"Nutrient deficiency may result in decreased plant productivity and/or plant fertility (McDowell et al., 2008; Morgan and Connolly, 2013; Stenberg and Muola, 2017)."*

5. L75-77, suggest changing "have to" to "can" and keeping fewer citations. The usage of self-adjustment is misleading in such context.

Reply: Done. The sentence has been revised and moved to Section 2.2.2 as follows:

*New Line 175:* *"Evidence has shown that plant CNR can change with nutrient availability (Chen and Chen, 2021; McGroddy et al., 2004; Meyer-Grünefeldt et al., 2015; Sardans et al., 2012; Smith, 1991;)"*

6. L77-79, please revise this sentence. Lipid is not a polymer. It should be "nutrient-starved". Unclear if the authors mean C/N ratios are influenced by being exposed to high light or the accumulation of C polymer are greater when exposed to high light. Please revise the term "high light".

Reply: The sentence has been revised and moved to Section 2.2.2 as follows:

*New Line 176:* *"Plant cell CNRs are influenced by the accumulation of C polymers, such as carbohydrates, and are greater when cells are nutrient starved or exposed to high levels of photosynthetically active radiation (PAR) (Aber et al., 2003; MacDonald et al., 2002; Talmy et al., 2014)."*

7. L82, suggest clarifying what the N is, such as soil N availability or plant N, and whether it is photosynthesis capacity or actual photosynthesis rate.

Reply: This sentence has been deleted in the revised version because it is redundant with other parts of the paper.

8. L96-99, please add references for the flux data and satellite-derived observational data.

Reply: Done.

*New Lines 86-91:* *"The coupled model is verified at thirteen flux tower sites (Lund et al., 2012; Pastorello et al., 2020) with different PFTs and is used to conduct several sets of global 2-D offline simulations from 1948 to 2007 to assess the effects of the coupling process. Model predictions of global GPP and LAI have been evaluated against satellite-derived observational data (Jung et al., 2009, Sheffield et al., 2006, Zhu et al., 2013). The results demonstrate the relative importance of different plant N processes in this C-N framework."*

9. L116, please clarify what "vegetation conditions" are. It should be "physiological".

Reply: The sentence has been revised to make it more specific.

*New Lines 106-107:* *"Moreover, the surface albedo and aerodynamic resistances are also updated based on the vegetation leaf area index, vegetation cover, vegetation height, and greenness."*

10. L119, suggest changing to "C4 grasses" to be precise and consistent with Tables 2 and 3. It is "tundra shrub" in Tables 1 and 2. Please clarify what is "net C availability".

Reply: Done. We have revised Tables 1 and 2 to maintain consistency. The term "Net C availability" was replaced with "NPP".

11. L125, consider listing the pools in a table to present the information more clearly.

Reply: We listed the nitrogen pools as suggested (also as new Table 1 in the paper).

**Table R2.** The Nitrogen Pools in DayCent-SOM

|  |  | **Aboveground** | **Belowground** |
|---|---|---|---|
| **Mineral N pool** |  |  | Soil mineral N pools |
| **Organic N pool** | non-woody litter pools | Surface structural N  Surface metabolic N | Soil structural N  Soil metabolic N |
|  | woody debris pool | Surface dead N |  |
|  | kinetically defined organic matter pools | Surface active N  Surface slow organic N | Soil active organic N  Soil slow organic N  Soil passive organic N |

12.  L134, please either clarity the temperature and moisture effects or remove the word "effects".

Reply: These words have been removed. The sentence has been revised as follows:

*New Line 125: "Each type of organic pool has its own intrinsic rate of decomposition, modified by temperature and moisture  (Parton et al., 1994)."*

13.  L140, please revise "plant life activities".

Reply: The term "plant life activity" was vague and has been deleted throughout the paper.

*New Line 140: "To represent C and N interactions, we have developed a plant C-N interface framework to couple biophysical and biochemical processes in the caron and nitrogen cycles ."*

14.  L141, L144, please refrain from using "/" excessively. Suggest changing "physical/biological" to "physical and biological" and "temperature/moisture" to "temperature and moisture". Please check for other "/" as well.

Reply: Done. Thank you for pointing this out. We eliminated many instances of "/" in the paper.

15.  L145, please revise this sentence and clarify "surface water", "carbon fluxes" (it is not mentioned in the second half of the sentence), and "plant litter" (e.g., as in fluxes for production and decomposition or pools).

Reply: Done. These sentences have been revised as follows:

*New Lines 144-146: "The soil N dynamics model (DayCent-SOM) is directly driven by soil temperature, soil moisture, net radiation and plant C and N litter inputs into the soil organic pool, which are provided by the SSiB5/TRIFFID. *

16.  L148-150, please revise this sentence. It is unclear what the authors mean by "N effects on plant physiology from photosynthesis, ... plus a dynamic C/N ratio".

Reply: The sentence has been revised as follows:

*New Lines 148-150:* *"Following plant N uptake from DayCent-SOM, our plant C-N interface framework describes the effects of N on photosynthesis, plant autotrophic respiration, and plant phenology (Fig. 1). All these effects are associated with a dynamic CNR."*

17. L152-154, please revise this sentence. It reads repetitive with "not only considers N limitation ... but also emphasizes the N limitation effect ..." and "help us obtain more information to understand ..."

Reply: The sentence has been revised as follows:

*New Lines 169-171* *"With consideration of the effect on phenology, the N limitation effect during the growth season is emphasized. All these considerations in the framework should help to understand the effects of N processes to the C cycle more comprehensively."*

18. L163-164, please revise the sentence to clarify the potential confusion that GPP follows autotropic respiration. Please revise "in plant life".

Reply: These sentences have similar meanings to those of other sentences and have been deleted from the revised paper.

19. L166, this might be controversial as in plants can certainly respond and adapt to lower N availability but it would be a stretch to certainly call it "adjust resource

Reply: *New Line 180*. We changed "adjusting resources" to "respond and adapt to lower N availability".

20. L167-170, please revise this sentence. It reads contradicting with "resorb only 50%" and "cause a major internal nutrient flux".

Reply: "A major internal nutrient flux" has been eliminated. The sentences have been revised to

*New Lines 181-184*. *"Studies show that plants resorb only about 50% of leaf N on average (Aerts, 1996) to conserve nutrients (Clarkson and Hanson, 1980) and to increase nutrient use efficiency (Herbert & Fownes, 1999; Vitousek, 1982). These processes cause changes in the CNR to reduce the impact of N limitation (Talhelm et al., 2011; Vicca et al., 2012)."*

21. L170-172, please revise this sentence to improve clarity and avoid going in circle, such as what affect plant productivity and litter N content. Now it reads like "plant responses affect plant productivity and litter N content".

Reply: This sentence has been deleted. This sentence tries to point out the fixed C/N ratio's shortcoming. However, actually, it did not provide real substance.

22. L173-174, please revise this sentence to clarify "improve plant responses".

Reply: This sentence has been eliminated from the revised paper to avoid repetition.

23. L179-196, the majority of this paragraph should fit in the introduction or discussion better.

Reply: This paragraph has been moved to Section 2.2.3 to provide background information for our parameterization of the N limitation effect on photosynthesis.

24. L197, please revise this sentence to increase clarity. For instance, NPP is part of the terrestrial C cycle.

Reply: *New Line 162*: The sentence was revised to *"Nitrogen is not the only dominant regulator of photosynthesis and vegetation dynamics"*.

25. L199, please clarify "normal N concentration".

Reply: In *the new Line 164*, we changed it to "In common N concentration range".

26. L203, please revise the sentence "Because plants need time to turnover, the plant N processes ..." for clarity and accuracy.

Reply: To reduce the number of repetitions, this and some other sentences have been removed.

27. L205, perhaps the authors mean "modulates LAI evolution, e.g., via leaf mortality?" Should it be "supplies" instead of "supplements?"

Reply: To reduce the number of repetitions, this and some other sentences have been removed.

28. L209, since C/N ratios is abbreviated as CNRs from here, why not introducing it from the start?

Reply: This is a good suggestion. We now apply the CNR from the start.

29. Regrettably, similar issues persist throughout the rest of the text. I will refrain from detailing them further until the authors have thoroughly revised the manuscript.

Reply: Please see our response to the major concern. The paper has reorganized to address this issue.

Overall, the general structure, clarity, terminology, as well as accuracy throughout the manuscript need to be substantially improved.

---

## Author Comment (AC2)

**Response to Anonymous Referee #2**

This study proposed a plant carbon-nitrogen coupling framework to improve a biophysical-ecosystem-biogeochemical model. The author ran the modified model at the site and global levels, and compared the model results with in-situ observations and remote sensing/machine learning estimations . Moreover, the authors conducted a series of experimental experiments at the global level to quantify the major effects of the N process and C-N interface coupling methodology on the C cycle. This study proposes a new approach, and considers the N limitation effects not only on photosynthesis but also on plant respiration and phenology. However, there are several significant drawbacks in this study. The reviewer has the following concerns and suggestions for the authors to consider:

Reply: Thank you very much for your comprehensive and constructive reviews. We appreciate your effort and acknowledge your review in the paper's "Acknowledgment".

Does the SSiB5/Triffid/DayCent-SOM v1.0 model consider anthropogenic N inputs (N deposition, fertilizer and manure) into terrestrial ecosystems? I guess no, since there is not no such information mentioned in the manuscript. If the model doesn't consider anthropogenic N inputs, the reported N limitation effects may be largely exaggerated because anthropogenic N inputs to terrestrial ecosystems are much larger than the vegetation N fixation in recent decades which can relief N limitation. In Figure 8 (f), the effect of N limitation is large in Eastern China and central USA, however, the anthropogenic N inputs were quite large in these regions (Tian et al., 2022), the N limitation shouldn't be large if anthropogenic N inputs are considered. This is my major concern.

Reply: The reviewer raised a very important point here. Our model includes anthropogenic N as a model input variable, and its impact is an important issue for investigation. In this paper, we did not address this issue. As a first paper for our C and N coupled model, the editor instructed us to focus on the description of model development in this resubmission. The reviewer's opinion regarding the anthropogenic effect has been well taken and included at the end of the revised paper as an important issue for further investigation (Tian *et al*., 2022).

*New Lines 619-623: "Anthropogenic N input is one of the major factors affecting C–*

*N coupling and N limitation. The anthropogenic N inputs to terrestrial ecosystems have been much greater than the vegetation N fixation in recent decades in some areas, such as eastern China and the central USA, which can relieve N limitations (Tian et al., 2022). Due to the scope of this paper, this issue is not addressed in this paper but is an important subject for further investigation to comprehensively understand the N limitation effect.*"

**Reference:**

Tian H, Bian Z, Shi H, *et al*. History of anthropogenic Nitrogen inputs (HaNi) to the terrestrial biosphere: a 5 arcmin resolution annual dataset from 1860 to 2019[J]. Earth System Science Data, 2022, 14(10): 4551-4568.

The SSiB5/Triffid/DayCent-SOM v1.0 model performs poor in modelling the magnitude of LAI although its performance is better than SsiB4. At the global level, SsiB5 estimation is about 100% higher than the remote sensing estimation (Figure 11)! Please elaborate on how is $LAI_{balance}$ calculated in model and the vegetation carbon allocation scheme. Also, it is necessary to add one paragraph discussing the potential reasons for the overestimation of LAI and the future improvement measures.

Reply: The reviewer points out an important shortcoming in the model's LAI simulation. Recent review papers confirm that the overestimation of C sequestration and LAI is a common issue in current dynamic vegetation models (Anav *et al*., 2013; Murray-Tortarolo *et al*., 2013; Zaehle *et al*., 2015; Mueller *et al*., 2019; Gristina *et al*., 2020; Oliveira *et al*., 2021; Heikkinen *et al*., 2021).

Murray-Tortarolo *et al*. (2013) and Anav *et al*. (2013) evaluated the performance of dynamic vegetation models in simulating LAI from a CMIP model intercomparison. A figure from their paper is attached below for your reference. Based on the figure, it is clear that this issue exists in most dynamic vegetation models. More recent papers, such as those cited above, also confirm this shortcoming in current dynamic vegetation models. It is important to overcome such large bias. In fact, this is one of the main motivations for us to introduce the N limitation into the Earth System Model. However, despite proper simulation of GPP after introducing N limitation, our results indicate that further efforts are still needed to improve LAI simulation. In the revised paper, we note that this is one of several issues that deserves further investigation.

Since overestimating LAI is a common problem in dynamic vegetation modeling, we only indicate that this is an issue that needs to be further investigated but did not elaborate this issue further. To understand how $LAI_{balance}$ is calculated, it needs a

substantial effort (not just a couple of paragraphs), which may distract the paper's main focus. Moreover, we are not sure whether LAI$_{balance}$ is the cause of the LAI overestimation. Nevertheless, we add references after the LAI$_{balance}$ in the revised paper for additional information.

In the ***New Lines 613-618***, we added a paragraph to address this issue.

*"Moreover, although the global GPP of SSiB5 was similar to that of the satellite-derived GPP, the positive bias for the LAI was still very large (Table 7). Recent review papers seem to confirm that overestimation of LAI is a common issue in current dynamic vegetation models. Murray-Tortarolo et al. (2013) and Anav et al. (2013) evaluated the performance of dynamic vegetation models in simulating LAI from a CMIP model intercomparison. The simulated LAI for almost every dynamic vegetation model is twice as large as the satellite-derived LAI. More recent studies (Zaehle et al., 2015; Mueller et al., 2019; Gristina et al., 2020; Oliveira et al., 2021; Heikkinen et al., 2021) have confirmed this shortcoming in current dynamic vegetation models. Further investigations are necessary."*

**Figure 2.** Linear trend against average LAI for each model and satellite observations, with IAV represented as colors. The data represents the whole high-latitude Northern Hemisphere (30°–90°) for the time period 1986–2005.

[Figure]

Murray-Tortarolo (2013)

**References:**

Anav, A.; Murray-Tortarolo, G.; Friedlingstein, P.; Sitch, S.; Piao, S.; Zhu, Z. Evaluation of land surface models in reproducing satellite Derived leaf area index over the high-latitude northern hemisphere. Part II: Earth system models. Remote Sens. 2013, 5, 3637–3661

Oliveira D. C. *et al*., Depth assessed and upscaling of single case studies might overestimate the role of C sequestration by pastures in the commitments of Brazil's low-carbon agriculture plan. Carbon Management. 12, 499–508.

Oliveira, D. C. de, Oliveira, D. M. da S., Freitas, R. de C. A. de, Barreto, M. S., Almeida, R. E. M. de, Batista, R. B., & Cerri, C. E. P. Depth assessed and upscaling of single case studies might overestimate the role of C sequestration by pastures in the commitments of Brazil's low-carbon agriculture plan. Carbon Management, 12(5), 499–508 (2021). https://doi.org/10.1080/1758300 4.2021.1977390

Heikkinen, J., Keskinen, R., Regina, K., Honkanen, H., & Nuutinen, V.. Estimation of carbon stocks in boreal cropland soils - methodological considerations. European Journal of Soil Science, 72(2), 934–945. (2021) https://doi.org/10.1111/ejss.13033

Gristina, L., Scalenghe, R., García-Díaz, A., Matranga, M. G., Ferraro, V., Guaitoli, F., & Novara, A.. Soil organic carbon stocks under recommended management practices in different soils of semiarid vineyards. Land Degradation and Development, 31(15), 1906–1914 (2020). https://doi.org/10.1002/ldr.3339

Murray-Tortarolo, G., Anav, A., Friedlingstein, P., Sitch, S., Piao, S.L., Zhu, Z. C., Poulter, B., Zaehle, S., Ahlstrom, A., Lomas, M., Levis, S., Viovy, N., and Zeng, N.: Evaluation of Land Surface Models in Reproducing Satellite-Derived LAI over the High-Latitude Northern Hemisphere. Part I: Uncoupled DGVMs, Remote Sensing, 5, 4819–4838, 2013.

Mueller, P., Ladiges, N., Jack, A., Schmiedl, G., Kutzbach, L., Jensen, K., & Nolte, S.. Assessing the long-term carbon-sequestration potential of the seminatural salt marshes in the European Wadden Sea. Ecosphere, 10(1). (2019) https://doi.org/10.1002/ecs2.2556

Zaehle, S., C. D. Jones, B. Houlton, J. F. Lamarque, E. Robertson, Nitrogen availability reduces CMIP5 projections of twenty-first-century land carbon uptake. Journal of Climate. 28, 2494–2511 (2015).

There is no tundra site in site-level validation. I recommend adding at least one tundra site. Please elaborate on the calculation of PFT fractional coverage in model, and add one figure comparing model results with satellite-based land cover product to justify that model can accurately estimate PFT fractional coverage.

Reply: Thank you for the suggestion. Our validation sites were limited to the AmeriFlux sites. We now include a tundra site (Lund *et al*., 2012) for validation. The figure attached below is included in Section 4.1, and the statistics for this site are included in Table 6 (previous Table 5). The results from the new 13-site average are consistent with the previous results with the 12-site average, as shown in the revised Table 6 (previous Table 5).

**Table R1.** Tundra site information used for model validation.

| Site name | LAT | LONG | PFT | Time |
|---|---|---|---|---|
| Zackenberg Heath | 74.47 | -20.55 | tundra | 2000-2014 |

[Figure]

**Figure R1**. Simulated seasonal variations in GPP, sensible heat, and latent heat against observations at the tundra site.

As to the simulated PFT distribution issue, we had two publications (Zhang et al., 2015;

Liu et al., 2019) extensively discuss our model's simulation of the global PFT distribution and fraction coverage and compare with the satellite derived map. The simulation results are generally consistent with observation (see figure below). The SSiB5/ TRIFFID/DayCent-SOM did not produce substantial difference in the PFT distribution with a few decades of simulation.

In the revised paper, in *new lines 494-495*, we add the following sentences: *"The SSiB4/TRIFFID-simulated global PFT distribution has been extensively discussed in Zhang et al. (2015) and Liu et al. (2019). The simulation results are generally consistent with observation. The spatial distribution from the SSiB5/TRIFFID/ DayCent-SOM did not show substantial difference and will not be discussed here."*.

[Figure]

[Figure]

**Figure 3.** Dominant vegetation type comparison between **(a)** GLC2000 and **(b)** SSiB4/TRIFFID, and **(c)** region definitions.

**References:**

Liu, Y., Xue, Y., Macdonald, G., Cox, P., and Zhang, Z.: Global vegetation variability and its response to elevated CO 2 , global warming, and climate variability - A study using the offline SSiB4/TRIFFID model and satellite data, Earth Syst. Dyn., 10, 9–29, https://doi.org/10.5194/esd-10-9-2019, 2019.

Lund, M., Falk, J. M., Friborg, T., Mbufong, H. N., Sigsgaard, C., Soegaard, H. and Tamstorf, M. P.: Trends in CO2 exchange in a high Arctic tundra heath, 2000–2010, J. Geophys. Res., 117(G2), G02001, 2012.

Zhang, Z., Xue, Y., MacDonald, G., Cox, P. M., and Collatz, G. J.: Investigation of North American vegetation variability under recent climate: A study using the SSiB4/TRIFFID biophysical/dynamic vegetation model, J. Geophys. Res., 120, 1300–1321, https://doi.org/10.1002/2014JD021963, 2015.

I suggest list equations that calculate key processes and variables in carbon and nitrogen cycles such as GPP, SOC/SON decomposition, plant N fixation, plant N uptake, and N mineralization.

Reply: The reviewer suggested listing the major equations for the coupled model. This is a very good suggestion that should help readers understand the results. However, SSiB5, TRIFFID, and DayCent-SOM are process-based models that involve numerous equations to obtain variables such as GPP and decomposition. After several attempts, we realize that it is difficult to select a proper set of equations to provide brief and useful information for readers to have a basic understanding of the major physical, biophysical, and ecological processes in the model. A handbook is needed to accomplish this task. We apologize that we had difficulty accomplishing this task. To have a very basic understanding as a starting point, we suggest reading Zhan *et al*. (2003, Ecological modeling), Cox (2001, Hadley Tech note), and Parton (1994, a Textbook, see reference in the paper).

The manuscript needs modifications on the structure. From my point of view, it is better to move line 164-176 and line 179-191 to the Introduction part, and the order of section 3.3 and 3.2 should be reversed.

Reply: Thank you for your constructive comments and suggestions. We agree that the paper structure needs to be improved. Per your and another reviewer's suggestion, we have rearranged parts of the Introduction and Sections (2.2.1-2.2.5), which describe the model development. For lines 164-176, we have moved to "Section 2.2.2 Dynamic C/N ratio based on plant growth and soil nitrogen storage" to provide background information on why we need a dynamic C/N ratio and why we parameterize the C/N ratio this way. This will provide better presentation flow and avoid repeating (i.e., similar things appear in both Introduction and relevant sections). Similarly, we have moved lines 179-191 to Section 2.2.3 to provide background information for our parameterization of the N limitation effect on photosynthesis.

In the order of Sections 3.2 and 3.3, the following is the reason why we present Section 3.2 first. In model development, introducing a realistic process does not necessarily improve the results due to model deficiencies. Validation is necessary to confirm the model's reliability. In Section 3.2., we demonstrate that after introducing a very complex N-processing model and N limitation effect, compared with the site measurement data, the original ability of the SSiB5/TRIFFID model to simulate seasonal and interannual variability in heat fluxes is intact, even with slight

improvement. This provides some confidence for our next long-term 2-D simulation presented in Section 3.3.

Moreover, more discussions on the limitations of the SSiB5/Triffid/DayCent-SOM v1.0 model and potential future developments are needed.

Reply: As discussed earlier, we noted the anthropogenic N input and large LAI bias issues for further improvement at the end of the paper. In addition, we also note the limitations of the offline simulation.

Please show some results of NlResp and NlPhen, otherwise, you should delete the descriptions of these experiments.

Reply: Thank you for this comment. We have discussed the effects of NIResp and NIPhen, but the previous presentation in the text is unclear. We conducted four experiments, namely, the NIResp, NIPhen, NIPSN, and SSiB5 experiments, in this research. Exp. SSiB5 showed a total effect, and another three experiments tested the effect of individual processes. However, only Exp. NIPSN and Exp. SSiB5 showed statistically significant results. Therefore, we mainly show the NIPSN and SSiB5 results individually, not the NIResp and NIPhen. However, the sum of these two effects also has a substantial effect on many parts of the world. Instead of showing individual results, we present the sum of these two effects. In Fig. 13b, we added a subtitle indicating that the figure shows NIPhen + NIPesp effects. In the new lines 586-590, we also added a much clearer discussion on the effects of NIResp and NIResp.

*"The results from Exp. NlResp or Exp. NlPhen individually did not show a statistically significant impact. However, the sum of these two N limitations still has substantial impacts on many parts of the world, as displayed in Fig. 13b, mainly in tropical rainforests and some midlatitude regions. In addition, the differences between Exp. SSiB5, which includes three limitations, and Exp. NIPSN, as displayed in Figs. 10 and 11, also delineate the characteristics of the global impacts of these two effects at seasonal and interannual scales."*

**Line-specific comments and suggestions:**

Line 86: Please list these plant N metabolism processes.

Reply:"Metabolism" is not a proper word here. This sentence, however, has been

deleted from the revised paper.

Line 104: 1982-2007 rather than 1948-2007.

Reply: Thank you for your careful review. We have corrected this.

Line 126: eight types rather than six types?

Reply: There were six pools in DayCent-SOM rather than eight. We listed the six nitrogen pools here for clarity (also as new Table 1 in the paper).

**Table R2.** The Nitrogen Pools in DayCent-SOM

|  |  | Aboveground | Belowground |
|---|---|---|---|
| **Mineral N pool** |  |  | Soil mineral N pools |
| **Organic N pool** | nonwoody litter pools | Surface structural N
Surface metabolic N | Soil structural N
Soil metabolic N |
|  | woody debris pool | Surface dead N |  |
|  | kinetically defined organic matter pools | Surface active N
Surface slow organic N | Soil active organic N
Soil slow organic N
Soil passive organic N |

Line 197: delete " and terrestrial CX cycles"

Reply: The sentence was modified in the new line 162 as follows: *"Nitrogen is not the only dominant regulator of photosynthesis and vegetation dynamics"*.

Lines 268-270: I suggest delete line 268-270 to avoid misinterpretation

Reply: Done. These lines have been deleted.

Line 284: I didn't find the paper: Yang et al., 1992

Reply: We added this paper to the References section.

Line 320: temporal resolution of vegetation dynamics is ten-day, is it too coarse for phenology (especially for the boreal forests and tundra)?

Reply: Many dynamic vegetation models use much longer time steps, such as 1 month and 1 year. For instance, the Orchidee model (as shown in Fig. 2 above) uses a 1-year time step. In SSiB5/TRIFFID/DayCent-SOM, SSiB5 provides GPP, autotrophic

respiration, and other physical variables, such as canopy and soil temperatures and soil moisture, every 3 hours for TRIFFID. However, TRIFFID accumulates the GPP from SSiB5 and produces biotic C, PFT fractional coverage, vegetation height, and LAI every ten days, which are then used to update surface properties in SSiB5, such as albedo, surface roughness length, and aerodynamic resistances. Our model's time step is relatively shorter than that of many other dynamic vegetation models. The ten-day accumulation of TRIFFID occurred because, if the time step is too short, changes in vegetation conditions may be even smaller than the noise, which may cause computational instability. We performed sensitivity tests with a 5-day time step, and the results were similar. Therefore, in this study, we retained a 10-day time step to save computer time. When applying this model for fire studies, we may have to use shorter time steps.

Line 395: How do you set up the equilibrium rum at the site level? The same with global run?

Reply: Yes. The equilibrium run at the site level is the same as the global run. We added one sentence to the revised paper to clarify this.

*New Line 404:* *"This approach was applied for both site and global 2-D simulation".*

Line 405: Four sets of sensitivity experiments rather than six sets?

Reply: Thank you for your careful review. We have corrected this.

Figure 7: SSiB5 is higher than in (g) and (k). In these two sites, SSiB5 has lower GPP than SSiB4, why its evapotranspiration (latent heat) is higher? This doesn't seem to make sense.

Reply: Thank you for your careful review. There are three components in our model that contribute to the total latent heat flux, as shown in Fig. 7. These factors include transpiration from the canopy, direct evaporation from the leaf due to interception loss of precipitation, and soil evaporation. In the offline test, the atmospheric demand is fixed; when transpiration (and GPP in general) is reduced/increased, soil evaporation must change to satisfy the atmospheric demand. This change is not linear because the sensible heat flux also changes. As such, the latent heat change in very few cases may not be consistent with the change in GPP due to the change in soil evaporation. However, for the 13-site average, the changes in GPP and total evapotranspiration are still consistent.

Figure 9: IS LAI the mean value of GIMMIS and GLASS?

Reply: We used the GIMMS LAI in this figure and added a note as follows.

*New Line 509:* *"Note: OBS in LAI is GIMMS LAI."*

Line 542-543: Is there any observational evidence for this vegetation transition?

Reply: To our knowledge, there is no observational evidence for this vegetation transition.

---

## Author Response (AR2)

**General comments:**

The authors successfully addressed some of the issues raised in the last round of review and have substantially improved the structure as well as the language of the manuscript. Although I believe the manuscript can still benefit greatly from carefully revising the text for readability, logic, and consistency.

Reply: Thank you very much for your encouragement and comments.

**Main concerns:**

- Regarding some of the main concerns for the previous manuscript version:

1.***"The framework implemented in the study revolves around processes in terrestrial N cycles, more specifically about plant N demand and stress. However, the relevant processes are overly simplified when describing the necessity to modify current representations in the models."*** In introduction, there is still the same issue that too much text is focused on the disadvantage of C-only models, while the focus should be the limitation of C-N models with fixed CNR. Especially that after talking about ecosystem N processes that regulate C cycles, it circled back to only a small portion of CMIP6 models include N cycles. As such, the paragraph mainly stresses out that C-only models cannot account for N limitations being problematic in potential overestimation of terrestrial C sequestration,

whereas the performance of C-N models with fixed CNR is not mentioned at all. This creates an obvious logic gap between "C-only models does not consider N limitations" and "this new coupling framework reduces the impact of N limitations which is an advantage (e.g., lines 775ff)". Why does the impact of N limitations need to be reduced in the first place?

**Reply:** Thank you for the comments/suggestions!

We would like to clarify here:

(1). How to properly introduce the N process to the C-only model is still outstanding research, as demonstrated in recently published papers, such as Davies-Barnard et al., 2020 and Kou-Giesbrecht et al. 2023, which claim that "a disconnect between the carbon and nitrogen cycles" is a major issue in current terrestrial biosphere modeling.

(2). The objective of this study is to improve C–N coupling in the Earth System Model, as indicated in the paper's introduction: *"presents a recently developed process-based plant C–N coupling framework with a consistent coupling strategy between biophysical and biogeochemical processes. The framework mainly focuses on the effects of N limitation on plant photosynthesis (Section 2.2.3), plant respiration (Section 2.2.4), and plant phenology (Section 2.2.5) with a dynamic C/N ratio (CNR, Section 2.2.2). The dynamic plant CNR is a more realistic representation than the fixed plant CNR in assessing the effect of N limitation on plant C processes and interactions between plant C and N processes."* (***new lines 69-73***)

As such, "reducing the impact of N limitation" is not a subject of this study. The quoted word "advantage" in line 775ff is not used for "reduced N limitation effect" but refers to our approach "does not linearly or instantaneously respond to the available N content". Sorry, our statements in lines 775ff caused confusion. We have revised this paragraph in the "Summary" section (previous Discussion and Conclusion section) to avoid confusion. "Advantage" is a subjective assessment. To avoid controversy, we deleted the sentence with "advantage".

(3). In Kou-Giesbrecht et al. (2023) and other recent C-N model intercomparison papers, various configurations in these coupled models are listed (see the attached tables below). Among the many coupling components, one item is whether to use the use of a fixed CNR or flexible CNR. These papers did not conclude whether "fixed" or "flexible" CNRs should be used or whether models with flexible CNRs are required to test fixed CNRs first. In our paper, we emphasize that the flexible CNR approach more realistically represents the ecological process. It is not our task to prove that a model with a fixed CNR cannot properly simulate coupling. To test the fixed CNR in our model, it is not simple to set the CNR as a constant but must change many other parameterizations, which actually requires setting a new coupling framework and is beyond the scope of this paper.

(4). The reviewer questions why "the impact of N limitations need to be

reduced in the first place". As discussed above, this is never a subject in our study. Our goal is to introduce a more realistic C–N coupling process. In this paper, we indicate that a flexible CNR reduces the N limitation effect compared with a fixed CNR. This finding does not mean our goal is to reduce the N-limitation effect through flexible CNR.

Another reviewer of this paper suggested that we add N deposition in future studies, which will reduce the N limitation effect in some areas. Apparently, that reviewer's suggestion is to include a more realistic process; reducing the N limitation is not the purpose of adding N deposition.

To avoid any confusion, we have deleted "reduce the N-limitation effect" in the paragraph in the Summary section. The paragraph has been revised as follows (***new lines 634-643***):

*"The new C-N coupling framework takes a consistent coupling strategy between biophysical and biogeochemical processes and mainly focuses on the effects of N limitation on plant photosynthesis, plant respiration, and plant phenology. The dynamic plant CNR is used to represent plant resistance and response to N stress, which allows adaptations in the stoichiometry of C and N. This approach increases nutrient use efficiency and takes into account N remobilization and resorption; the N limitation effect does not linearly or instantaneously respond to the available N content. A linear relationship between the N limitation factor and available N is valid only when N availability is not sufficient for the minimum N*

*demand for new growth.”*

**Table 1.** Key nitrogen cycle algorithms applied by the models. C is Carbon; N is Nitrogen; GPP is gross primary productivity; NPP is net primary productivity; and PFT is plant functional type.

| | CLM4.5 | CLM5 | JSBACH | JULES-ES | LPJ-GUESS |
|---|---|---|---|---|---|
| Key references | Oleson et al. (2010) | Lawrence et al. (2019) | Goll et al. (2017), Mauritsen et al. (2019) | Wiltshire et al. (2020) | Smith et al. (2014) |
| N effect on GPP | Downregulation of GPP to match stoichiometric constraint from allocable N | Leaf N compartmentalised into different pools to co-regulate photosynthesis according to the LUNA model | No direct effect | No direct effect | Reduction of Rubisco capacity in the case of N stress |
| N effect on autotrophic respiration | N content-dependent tissue-level maintenance respiration | Updated PFT-specific N-dependent leaf respiration scheme | No direct effect | N content-dependent maintenance respiration for roots and stems | N content-dependent maintenance respiration for roots and stems; leaf respiration reduced under N stress |
| Vegetation pool C : N stoichiometry | Fixed for all pools | Flexible for all pools | Fixed for all pools except labile | Flexible leaf stoichiometry from which root and stem C : N are scaled with fixed fractions | Flexible for leaves and fine roots; fixed otherwise |
| Retranslocation of N from shed leaves | Fraction of leaf N moved to mobile plant N pool prior to shedding; fraction depends on PFT-specific fixed live leaf and leaf litter C : N ratios | Fraction of leaf N moved to mobile plant N prior to shedding via two pathways: a free retranslocation or a paid-for retranslocation dependent on PFT-specific dynamic leaf C : N range and minimum leaf litter C : N as well as available carbon to spend for extraction in the FUN model | Fraction of leaf N moved to mobile plant N pool prior to shedding | Fraction of leaf N moved to labile store with PFT-specific re-translocation coefficient | Fraction of leaf N moved to mobile plant N pool prior to shedding; fraction depends on N stress |
| Biological N fixation | Monotonically increasing function of NPP | Symbiotic N fixation according to the FUN model; asymbiotic N fixation linearly dependent on evapotranspiration | Non-linear function of NPP | Linear function of NPP, 0.0016 kg N per kg C NPP | Linear function of ecosystem evapotranspiration, 0.102 mm yr$^{-1}$ ET + 0.524 per kg N ha$^{-1}$ yr$^{-1}$ |

Table in T. Davies-Barnard, et al. 2020

**Table 1.** Terrestrial biosphere models in the TRENDY-N ensemble and descriptions of their representations of N limitation of vegetation growth, biological N fixation, vegetation response to N limitation (i.e., strategies in which vegetation invests C to increase N supply in N-limited conditions), and N limitation of decomposition.

| | Reference | N limitation of vegetation growth | Biological N fixation | Vegetation response to N limitation | N limitation of decomposition |
|---|---|---|---|---|---|
| CABLE-POP | Haverd et al. (2018) | $V_{cmax} = f$(vegetation N) Flexible C : N stoichiometry | Time invariant | Static | N invariant |
| CLASSIC | Melton et al. (2020) | $V_{cmax} = f$(vegetation N) Flexible C : N stoichiometry | $f$(N limitation of vegetation growth) | Dynamic (biological N fixation) | N invariant |
| CLM5.0 | Lawrence et al. (2019) | $V_{cmax} = f$(vegetation N) Flexible C : N stoichiometry | $f$(N limitation of vegetation growth) | Dynamic (biological N fixation, mycorrhizae, re-translocation) | $f$(soil N) |
| DLEM | Tian et al. (2015) | GPP $= f$(vegetation N) | $f$(soil temperature, soil moisture, soil C, soil N) | Dynamic (root allocation) | $f$(soil N) |
| ISAM | Shu et al. (2020) | GPP $= f$(vegetation N) | $f$(ET) | Static | $f$(soil N) |
| JSBACH | Reick et al. (2021) | NPP $= f$(vegetation N) | $f$(NPP) | Static | $f$(soil N) |
| JULES-ES | Wiltshire et al. (2021) | NPP $= f$(vegetation N) | $f$(NPP) | Static | $f$(soil N) |
| LPJ-GUESS | Smith et al. (2014) | $V_{cmax} = f$(vegetation N) Flexible C : N stoichiometry | $f$(ET) | Dynamic (root allocation) | N invariant |
| LPX-Bern | Lienert and Joos (2018) | NPP $= f$(vegetation N) | Derived post hoc to simulate a closed N cycle | Static | N invariant |
| OCNv2 | Zaehle and Friend (2010) | $V_{cmax} = f$(vegetation N) Flexible C : N stoichiometry | $f$(N limitation of vegetation growth) | Dynamic (root allocation) | $f$(soil N) |
| ORCHIDEEv3 | Vuichard et al. (2019) | $V_{cmax} = f$(vegetation N) Flexible C : N stoichiometry | Time invariant | Static | N invariant |

Table in Kou-Giesbrecht, S., et al. 2023

**2.*"The need to evaluate plant C processes under the modified N processes is well motivated in the introduction. However, the connection between N processes and heat fluxes is absent"*,** I intended to remind adding some information on how ecosystem N processes interact with heat fluxes and why it is important to look at these variables (as how terrestrial C sink hinges on ecosystem N processes). It remains missing in the introduction and discussion as the first time "heat flux" is brought up is in Methods while being a main part of the Results. I would suggest adding a few sentences in the end of the introduction justifying the choice of all the variables.

**Reply:**

The reviewer raises a very important point here. We agree with the reviewer that the impact on heat flux is important! We de-emphasized this in this paper because the offline experiment did not have a significant impact on the heat flux. Per the reviewer's suggestion, we added a few sentences to justify our variable selection in this paper.

*New Lines 90-94: "In addition, the effects of N limitation on heat fluxes are also preliminary assessed with station data (Section 4.1). The results indicate that because the atmospheric forcings (such as downward radiation) in our offline experiment are the same for both the control and sensitivity runs, the heat flux response due to N limitation is limited. In this paper, we mainly focus on the GPP and LAI. A comprehensive assessment*

*of the effect of N limitation on heat fluxes and atmospheric circulation needs to be conducted in a fully coupled atmosphere–land model."*

3. It is nice to see how including more dynamic N processes mostly brings modelling results closer to the observations at global as well as site levels. However, it is curious that the amplitude of mean seasonality of GPP (Figure 10) is much dampened with NIPSN and SSiB5 compared to SSiB4 which seems closer to the observations. In this sense, instead of the claim of "improvement in the simulation of the seasonal cycle in SSiB5 (lines 672ff)", it only shows that mean monthly GPP is improved to different extents by months. This result should be explained potentially together with the changes in spatial patterns Figure 8 and Table 7. See the following studies on comparing modelling and observations for seasonality or seasonal biases of GPP:

- Lin S, Hu Z, Wang Y, Chen X, He B, Song Z, Sun S, Wu C, Zheng Y, Xia X, et al. 2023. Underestimated Interannual Variability of Terrestrial Vegetation Production by Terrestrial Ecosystem Models. Global Biogeochemical Cycles 37(4): e2023GB007696.

- MacBean N, Scott RL, Biederman JA, Peylin P, Kolb T, Litvak ME, Krishnan P, Meyers TP, Arora VK, Bastrikov V, et al. 2021. Dynamic global vegetation models underestimate net CO2 flux mean and inter-annual variability in dryland ecosystems. Environmental Research Letters

16(9)

**Reply:** Thank you for pointing out the seasonality issues.

After a more careful evaluation, we realized that averaging seasonality *globally* can be misleading due to the opposite seasonal patterns in the Northern (NH) and Southern Hemispheres (SH). Therefore, we have redrawn the seasonality separately for the NH and SH in Figures 10 and 11 and modified the discussion. In this analysis, we excluded high-latitude regions (50°N-60°N) due to less reliable satellite data records (Gonsamo et al., 2019) to ensure a more proper comparison. After those modifications, the simulated seasonality in our model runs showed a general consistency with the satellite products.

In *new lines 532-539*, we added the following discussion:

*"Furthermore, the interannual variability and annual cycle are also assessed. The correlation for interannual variability (Fig. 10a) in SSiB4 is already very high (0.98). SSiB5 continues keeping the high correlation as SSiB4. However, the standard deviations for the observations of SSiB4 and SSiB5 are 14.7, 26.7, and 19.9, respectively. SSiB5 is closer to the observations. The underestimation of interannual variability in terrestrial vegetation production by terrestrial ecosystem models (Lin et al., 2023: MacBean et al., 2021) does not appear in this study. The temporal correlation coefficients between the observed and simulated monthly mean GPPs for the Northern and Southern Hemispheres increased from*

*0.73/0.50 (Exp. SSiB4) to 0.75/0.55 (Exp. SSiB5), respectively (Figs. 10b and c), showing improvement in the simulation of the seasonal cycle in SSiB5".*

4. The shift in regional GPP biases to negative by SSiB5 shown in Table 7 requires more description and explanation in discussion which is largely omitted (e.g., lines 668ff, 786ff). It might be too much work at this point, however I wonder if it is possible to include some maps for NPP and autotropic aspiration (SSiB4 vs SSiB5) to show how to attribute the improvement in GPP. NPP and respiration (please specify autotropic, heterotrophic, or both) are also mentioned in the discussion without presenting any data (line 794). Although NlPSN, NlResp, and NlPhen are showing the effects of each process, the interactive effects on NPP and autotropic aspiration may provide some information on biases in spatial patterns and seasonality of GPP.

**Reply:**

The reviewer requests a more in-depth discussion on the spatial patterns and seasonality of GPP biases and the role that each process plays. At the end of the review (question for Lines 785ff), the reviewer also raises the significant test issue.

We apologize. In the previous version, the significance test was not clearly addressed. In the revised paper, in ***new lines 470-471***, we add a statement

that *"the improvement in the SSiB5 model bias compared to SSiB4 that are presented in Table 6, are all statistically significant at the α = 0.05 level of the t test values."*.   We also add note in Tables 7 and 8 for the results' statistical significant.

This paper mainly presents the climatological results from model development. Only Figures 10 and 11 very briefly show the corrected seasonality results. As indicated in the last review, per the editor's instructions, the current paper version is focused mainly on describing model development. The discussion on scientific issues is not the subject of this paper. As such, in this submission, we only include basic validation for model development (some discussion on scientific issues has been removed in this submission). This paper already has 13 figures plus 8 tables. Any comprehensive discussion on seasonality and more detailed roles for each process is beyond the scope of this paper.

Since the reviewer wants to see the NPP figure, we have attached the difference in the NPP between SSiB5 and SSiB4 for reference. This result is generally consistent with that difference in GPP. For respiration, as indicated in the last review that, although the differences between SSiB5 and SSiB4 and between NIPSN and SSiB4 are statistically significant. The differences between the total respiration effect and that of SSiB4 were not significant. Therefore, we only present the results for SSiB5-SSiB4, NIPSN-SSiB4, and (SSiB5-NIPSN)-SSiB4, which shows the effect of

transpiration plus phenology. The individual respiration results, therefore, are not discussed separately.

[Figure]

I think it is great that the discussion has been expanded to additional N input. However I found a conclusive statement missing towards the end of the manuscript.

**Reply:**

Thank you for the suggestion. This issue is rooted in the model structure. We now separate the "Discussion and Conclusion" to the "Discussion" section and the Summary section. We have added the following statement at the end of the Summary section.

*New Lines 656-661. "Although significant progress has been made in recent years in incorporating the N cycle and its effect on the C cycle in the terrestrial biosphere in a number of ESM LSMs (with various representations of N processes), our and other relevant studies suggest that*

*there are still many important outstanding issues, some of which were discussed in Section 5, and further efforts in improving terrestrial biosphere modeling that represents the coupled C–N cycle are imperative for realistic process representation (Davies-Barnard et al., 2020; Kou-Giesbrecht et al., 2023) to better simulate N/C/climate interactions and future projections. We hope our efforts presented in this paper can stimulate more effort to work in this direction."*

**Minor points:**

Regarding the number of coupled models in CMIP6 and models with N cycle, it is unclear and potentially misleading as "11 out of 112 models include N cycle" (lines 81ff). Please clarify if you are focusing on the land vegetation models and the portion of them with interactive N module.

**Reply:**

Thank you for your enquiry. We revised this part to clarify the statement.

***New Lines 62-68.*** *"In the latest Coupled Model Intercomparison Project Phase 6 (CMIP6, Eyring et al., 2016), although there were 112 different coupled ESMs with various land surface models from 33 institutions, only 6 ESMs that incorporated an N cycle module contributed to the CMIP6 model intercomparison study on carbon concentration and carbon–climate feedback (Arora et al., 2020). In CMIP5, there were only 2 ESMs with N cycle modules included in the same model intercomparison study (Arora et*

*al., 2013). The coupling of N processes in ESM is still an important area of model development (Ghimire et al., 2016; Yu et al., 2020)."*

Please note that in our last version, we only indicate "fewer than 10" models. We now provide a more precise number, which is 6.

If keeping results and discussion separated, please consider restraining from discussing results and referring to other studies in the result section.

**Reply:**

Thank you for this very good suggestion. Several changes have been made.

(1). In the "Results" section, we have deleted the sentence to discuss the effect of phosphorous.

(2). We separated the "Discussion and Conclusion" section into a discussion section and a summary section. In the summary section, we more concisely summarize our major results to avoid simple repeating. We also added a concluding paragraph to the end of the paper (to respond to the reviewer's comment above).

Thank you for adding the site comparison for tundra. In Figure R1, the flatline for GPP 2011-2012 seems to be connecting the missing data which should be corrected. Please check other figures as well, e.g., Figure 5m, Figure 6b, and several plots starting with flatlines.

**Reply:**

Thank you for your careful review. It is indeed that some site data are missing in Fluxnet 2015. We have replaced the flat lines with blanks.

[Figure]

[Figure]

[Figure]

Please revise the captions to be independent and self-explanatory instead of "same as Figure x…"

**Reply:** Done. We have deleted "same as Figure XXX" and revised these with independent and self-explanatory figure captions (Figures 6, 7, 11, and 12).

Please check all the table and figure captions and if they are referred to correctly (e.g., line 806 should be Table 8).

**Reply**:Done.

Please make sure all the abbr. are explained first, including the ones in tables.

**Reply**:Thank you for your careful review. We have checked and revised the manuscript to ensure that it is accurate and explained first (in *new lines 20, 37, 46, 59, and 106*).

Please be precise with terms such as "plant N processes" vs "ecosystem N processes", "simulation" vs "prediction" (not recommended), "Vmax" vs "Vcmax" vs "Vc, max" etc.

**Reply:** Thank you for your careful review. In the revised paper, "ecosystem N processes" has been replaced by "plant N processes". "Model prediction" has been replaced by "model simulation". "Vmax" and "Vcmax" have been replaced by "$V_{c, max}$".

Not all "C/N ratios" were replaced by "CNRs".

**Reply**:In the revised paper, all "C/N ratios" have been replaced with "CNR".

I suggest the authors again to restrain from citing excessively. Please select the most representative references carefully instead of accumulating all the

citations for a well-established or well-recognised statement. For instance, new lines 54ff: "Adequate C-N coupling in plant N processes has been indicated as an area that still needs intensive investigation (Thum et al., 2019; Ghimire et al., 2016; Goll et al., 2017; Yu et al., 2020; Zaehle et al., 2015; Zhu et al., 2019)" does not need all six citations to back up the need of the research (which is then repeated multiple times unnecessarily).

**Reply:** Thank you for your suggestion. We selected the most representative references as suggested.

*New Lines 66-68*: *"The current status of C-N coupled models in the CMIP model intercomparisons and knowledge gaps and divergent theories in C-N coupling parameterizations suggest coupling of N processes in ESM is still an important area of model development (Thum et al., 2019; Ghimire et al., 2016; Goll et al., 2017; Yu et al., 2020 Zaehle et al., 2015; Zhu et al., 2019)."*

**Line-specific comments (correspond to the pdf file with tracked changes):**

Throughout the manuscript, it remains common for the sentences with redundancy, lack of precision, unclear language, and logical inconsistency. For instance:

Lines 73ff: "Some key plant N processes, such as N limitation on GPP, the effect of biomass N content on autotrophic respiration, plant N uptake,

ecosystem N loss, and biological N fixation, have been introduced into LSMs with various complexities to determine the effects of N limitation in current land models", from biological N fixation as one source of N input into the ecosystem and excessive N for plant use leaving the ecosystem are not necessarily plant N processes; as the effect of N on autotrophic respiration is specified as biomass N content, what about impact of N limitation on GPP? Leaf N content? Implication of such processes in LSMs is not intended to determine the effects of N limitation on models, but on C-N cycles using models… Please revise and add citations.

**Reply:** We agree that this part is not closely associated with the text before and after. It is rather confusing. We have deleted this part and replaced it with *"Several parameterizations have been developed in LSMs with various complexities to determine the effects of N limitation"* (**new line 57**).

Lines 78ff: "These methods include, for instance, using N to scale down the photosynthesis parameter V(c, max) (Ghimire et al., 2016; Zaehle et al., 2015) or potential GPP to reflect N availability (Gerber et al., 2010; Oleson et al., 2013; Wang et al., 2010), defining the C cost of N uptake (Fisher et al., 2010) and optimizing N allocation for leaf processes (Ali et al., 2015)", do you mean using N availability or N stress to scale down Vcmax and potential GPP? It reads like suggesting N availability can be reflected by how Vcmax and potential GPP are scaled down by N (also, what N? Soil

N or plant N uptake?), which is a logic loop; do you mean the carbon cost for BNF by Fisher et al. 2010a? Please revise and specify.

**Reply:** We have modified this paragraph as follows:

*New Lines 58-61: "These methods include, for instance, using leaf N availability to scale down the photosynthesis parameter $V_{c,max}$ (Ghimire et al., 2016; Zaehle et al., 2015) or potential GPP  (Gerber et al., 2010; Oleson et al., 2013; Wang et al., 2010), defining the energetic cost of N uptake (Fisher et al., 2010) and optimizing N allocation for leaf processes (Ali et al., 2015)".*

The sentence for C cost is from Fisher et al.'s paper. Introducing BNF needs a lot of explanation plus BNF is not that closely associated with this paper, which would cause confusion.

Lines 81ff: "The wide variety of assumptions and formulations of N cycling processes and C-N coupling reflects knowledge gaps and divergent theories, and further investigation is imperative (Kou-Giesbrecht, S., et al. 2023)", "The coupling of N processes is still an area of model development", "In the latest Coupled Model Intercomparison Project Phase 6 (CMIP6, Eyring et al., 2016), although there were 112 different coupled models with various land surface models from 33 research teams, only 10 models incorporated an N cycle module (Arora et al., 2020)", and 54ff: "Adequate C-N coupling in plant N processes has been indicated

as an area that still needs intensive investigation" are repetitive. Please revise and rearrange.

**Reply:** The sentence "Adequate C-N coupling in plant N processes has been indicated as an area that still needs intensive investigation" was deleted, and revisions were made to improve the presentation flow. The paragraph has been modified as follows:

*New Lines 54-68: "The fundamental aspects of N cycling for terrestrial biosphere models, such as N limitation of vegetation growth, strategies in which vegetation invests C to increase the N supply under N-limited conditions, and N limitation of decomposition, have been identified as important challenges for representing N cycling in terrestrial biosphere models (Meyerholt et al., 2020; Peng et al., 2020; Zaehle et al., 2015). Several parameterizations have been developed in LSMs with various complexities to determine the effects of N limitation. These methods include, for instance, using leaf N to scale down the photosynthesis parameter $V_{c,max}$ (Ghimire et al., 2016; Zaehle et al., 2015) or potential GPP (Gerber et al., 2010; Oleson et al., 2013; Wang et al., 2010), defining the energetic cost of N uptake (Fisher et al., 2010) and optimizing N allocation for leaf processes (Ali et al., 2015). There are wide variety of assumptions and formulations of N cycling processes and C-N coupling in land models. Furthermore, in the latest Coupled Model Intercomparison Project Phase 6 (CMIP6, Eyring et al., 2016), although there were 112*

*different coupled ESMs with various land surface models from 33 institutions, only 6 ESMs that incorporated an N cycle module contributed to the CMIP6 model intercomparison study on carbon concentration and carbon–climate feedback (Arora et al., 2020). In CMIP5, there were only 2 ESMs with N cycle modules included in the same model intercomparison study (Arora et al., 2013). The current status of C-N coupled models in the CMIP model intercomparisons and knowledge gaps and divergent theories in C-N coupling parameterizations suggest coupling of N processes in ESM is still an important area of model development (Ghimire et al., 2016; Yu et al., 2020)."*

Table 1: I am not sure about "dead N".

**Reply:** "Dead N" refers to woody debris N pools generated from the death of large wood, fine branches, and coarse roots. A note has been added to Table 1.

Line 214: change the "to" to "on" in "effects of N processes to the C cycle".

**Reply:** Thank you. ***New Line 176*** has been revised to *"All these considerations in the framework should help to understand the effects of N processes  the C cycle more comprehensively"*.

Line 217: what do you mean by plant fertility and how does it differ from

plant productivity?

**Reply:** This sentence has been revised to *"Nutrient deficiency may result in decreased soil fertility and/or plant productivity."* (***new line 180***)

Figure 4: site names are difficult to read.

**Reply:** We have changed to lager font size in the figures to help readability.

Figure 9: the formatting marks are visible for the texts.

**Reply:** The formatting marks have been removed.

[Figure]

Tables 7 and 8: do MTE and GIMMS need a column for bias?

**Reply:** Thank you for your suggestion. The first column of bias was removed.

Lines 771ff: "This study presents improvements in modeling the C cycle by introducing plant N processes into SSiB5/TRIFFID/DayCent-SOM, using DayCent-SOM to obtain the amount of N available to plants and plant soil N uptake". Please clearly specify the improvement is only regarding the previous SSiB/Triffid model version.

**Reply:** This has been revised as follows:

*New Line 634: "This study presents improvements in modeling the C cycle, compared to that of SSiB4/TRIFFID, by introducing plant N processes specifically into SSiB5/TRIFFID/DayCent-SOM. The DayCent-SOM provides the amount of N available to plants and plant soil N uptake."*

Lines 774ff: please specify that the dynamic CNR is for different PFTs (e.g., not for soil) and to what the plant resistance and responses are referred to (e.g., N stress).

**Reply:** The sentence has been changed as follows:

*New Lines 639-640: "The dynamic plant CNR is used to represent plant resistance and response to N stress, which allows adaptations in the stoichiometry of C and N."* In the next sentence of the original text, we

explained "resistance": *"This approach increases nutrient use efficiency and takes into account N remobilization and resorption; the N limitation effect does not necessarily linearly or instantaneously respond to the available N content"*.

Lines 775ff: just because "these processes can increase nutrient use efficiency and reduced the impact of N limitation" and "a linear relationship … is only valid when N availability is not sufficient for the minimum N demand for new growth", it is not clear to me how it is an advantage.

**Reply:** "Advantage" is a subjective assessment. To avoid controversy, we deleted the sentence with "advantage".

Line 780: I don't think "the state of plant growth" is used correctly here. It is also never mentioned elsewhere.

**Reply:** Thank you for pointing this out. The phrase "the state of plant growth" has been replaced with "N sufficiency". The sentence has been revised as follows:

*New Lines 643-644: "With the new model structure, the impacts of N on GPP are predicted directly but not linearly with leaf N content, which is affected by  N sufficiency, autotrophic maintenance and growth respiration, and plant phenology."*

Lines 782ff: it is questionable that "by comparing site-level results" can be evidence for "enhanced global model performance". Especially that only a few sites showed noticeably improved results compared to observations. Please revise.

**Reply:** The sentence has been changed to *"encourage us to carry out assessments of global performance"*.

Lines 785ff: "… produced significantly less absolute bias for GPP and LAI" is not tested statistically.

**Reply:** See our response to main concern 4.

**Comments from referee #2**

The authors have carefully revised the manuscript and addressed most of my suggestions. Since SSiB5/Triffid/DayCent-SOM v1.0 model has anthropogenic N inputs, the authors should introduce the N input data (fertilizer, manure, atmospheric deposition) used to drive the model. If N inputs data were not used to drive the model, the authors should write a paragraph to discuss that the reported N limitation effects may be largely exaggerated. This is an important issue and should be clearly stated. For LAI, according to my experience, most LSMs don't overestimate this much (100%), I still think it is important to discuss the potential improvement measures.

**Reply:** Per the reviewer's suggestion, we have extended our original discussion on the effect of anthropogenic N to one full paragraph.

*New Line 624-628:"Anthropogenic N input is one of the major factors affecting C–N coupling and N limitation. The anthropogenic N inputs to terrestrial ecosystems have been much greater than the vegetation N fixation in recent decades in some areas, such as eastern China and the central USA. As such, anthropogenic N input can relieve N limitations there (Tian et al., 2022). Due to the scope of this paper, we did not use anthropogenic N inputs to drive our model. This is an important issue for further investigations to comprehensively understand the effect of N limitation."*